# Efficient Personalized Federated Learning via Adaptive Weight Clustering Pruning

## Abstract

This paper introduces a novel personalized federated learning approach, Adaptive Federated Weight Clustering Pruning (AdFedWCP) (Rahaman et al., 2019), specifically designed to optimize communication efficiency in heterogeneous network environments. AdFedWCP innovatively combines adaptive weight clustering pruning techniques, effectively addressing data and bandwidth heterogeneity. By dynamically adjusting clustering centroids based on layer importance and client-specific data characteristics, it significantly reduces communication overhead. Experimental results demonstrate reductions in communication volume by up to 87.82% and accuracy improvements of 9.13% to 21.79% over baselines on EMNIST, CIFAR-10, and CIFAR-100. These findings underscore AdFedWCP's effectiveness in balancing communication efficiency and model accuracy, making it suitable for resource-constrained federated learning.

## 1 Introduction

The rapid growth of distributed data and rising data privacy concerns have made Federated Learning a promising paradigm for collaborative machine learning (McMahan et al., 2017). Federated learning enables multiple parties to train a shared model without exchanging raw data, ensuring user privacy and regulatory compliance (Li et al., 2023). Traditionally, federated learning relies on a central server to aggregate updates from clients into the global models (McMahan et al., 2017). However, due to data heterogeneity, single global models may not perform well across all clients, resulting in performance degradation and convergence challenges (Fallah et al., 2020). Personalized Federated Learning has emerged as a solution to this problem. The goal of personalized federated learning is to generate personalized models tailored to the local data of each client while retaining the advantages of the global models (Fallah et al., 2020). By introducing personalization, federated learning can effectively address the variations in data distribution across clients, improving model performance on individual clients. These personalized models can be implemented through methods such as knowledge distillation, model agnostic meta learning, or multi-task learning, allowing models to adapt to local data while retaining the benefits of the global models (Psaltis et al., 2023) (Fallah et al., 2020) (Marfoq et al., 2021).

Although personalized federated learning has been widely explored to address data heterogeneity, communication overhead remains a significant challenge. Communication overhead is a key factor in personalized federated learning as it affects training efficiency and system scalability (Kairouz et al., 2021). Bandwidth heterogeneity, a common challenge in real-world scenarios, refers to the variation in network bandwidth across clients due to differing network conditions and infrastructure capabilities (Kairouz et al., 2021) (Lim et al., 2020). Personalized federated learning methods such as FedEM (Marfoq et al., 2021), FedMask (Li et al., 2021), and pFedGate (Chen et al., 2023), despite making progress in handling client-specific needs, have not fully addressed the reduction of communication costs. FedMask and pFedGate reduce the size of parameter transmission through model pruning and gating layers, achieving an initial reduction in communication costs. However, these methods often rely on simple sparse parameter transmission without fully utilizing advanced compression techniques and lack flexibility in adapting to environments with significant client bandwidth differences and complex network conditions. Particularly in the case of bandwidth heterogeneity, existing methods cannot dynamically adjust compression strategies to optimize bandwidth resource usage, potentially resulting in communication bottlenecks in certain situations.

In conclusion, while these methods help with personalization, they face common limitations in communication efficiency and lack flexibility in handling bandwidth heterogeneity in different environ-

ments. Therefore, there is a need for an adaptive method that optimizes communication efficiency while maintaining model accuracy and dynamically adjusting to the needs of different clients. This issue raises a critical research question:

*How to design personalized federated learning methods that minimize communication cost without compromising model accuracy?*

To address these challenges, we propose AdFedWCP (Adaptive Federated Weight Clustering Pruning), which integrates adaptive weight clustering pruning with client-specific optimization. Our goal is to improve the efficiency and effectiveness of federated learning in bandwidth-constrained settings without compromising model performance. AdFedWCP is not a combination of existing techniques but introduces innovative strategies, such as weight clustering pruning, to uniquely address the distinct challenges of federated learning environments.

Our method directly addresses the issue of data heterogeneity through a global momentum-based update strategy, which enhances model convergence and generalization in a heterogeneous environment. At the same time, we use a saliency-based approach to assess the importance of each model layer, dynamically adjusting the number of cluster centroids per layer based on this importance, as well as client data characteristics and communication constraints.

The main contributions of this paper are as follows:

- A comprehensive Dynamic Weight Clustering Pruning Mechanism that reduces communication overhead and model size. This mechanism dynamically adjusts cluster centroids per layer based on layer importance, client data, and communication constraints through an integrated adaptive scheme, effectively addressing bandwidth heterogeneity.

- Comprehensive empirical evaluations demonstrate that AdFedWCP significantly reduces communication overhead and enhances overall performance across various datasets and network architectures. Specifically, AdFedWCP achieves a reduction in communication overhead of 87.54% to 87.82% on LeafCNN and LeNet network architectures, outperforming other baseline methods. In terms of accuracy, AdFedWCP achieves improvements ranging from 0.40% to 21.79% compared to baseline methods across datasets including EMNIST, CIFAR-10, and CIFAR-100. Overall, AdFedWCP improves communication efficiency by approximately 88% while maintaining comparable or superior accuracy, exhibiting significant advantages over all baseline methods.

## 2 RELATED WORK

### 2.1 PERSONALIZED FEDERATED LEARNING TO REDUCE COMMUNICATION OVERHEAD

Many methods aim to reduce the communication overhead in personalized federated learning, but have limitations in handling data heterogeneity or bandwidth heterogeneity. pFedGate reduces communication costs by generating personalized models through gating layers, but has difficulties in handling complex architectures and lacks flexibility to adapt to different bandwidth conditions, resulting in potential communication bottlenecks (Chen et al., 2023). FLuID dynamically adjusts the model size according to client resources to improve efficiency, but does not address the data heterogeneity problem (Wang et al., 2024). FedMask personalizes the model using client-specific pruning masks, which reduces communication costs, but cannot dynamically adapt to changing bandwidth conditions, resulting in potential inefficiency in environments with significant bandwidth heterogeneity (Li et al., 2021). PerAda integrates adapter modules and knowledge distillation to improve model generalization and communication efficiency, but similar to FedMask, it does not consider bandwidth heterogeneity, limiting its flexibility in varying network conditions (Xie et al., 2024).

### 2.2 PERSONALIZED FEDERATED LEARNING TO SOLVE DATA HETEROGENEITY

Other methods mainly target the data heterogeneity problem, but they are insufficient in optimizing the communication overhead. FedEM models client data as a mixture of distributions, effectively addressing the data heterogeneity problem, but its high communication and computational costs make it less suitable for bandwidth-constrained environments (Marfoq et al., 2021). AlignFed employs personalized feature extractors with a shared classifier to mitigate feature shifts and statistical differences, yet it does not fully eliminate distribution discrepancies (Zhu et al., 2024). TailorFL enhances personalization and resource efficiency through data-driven pruning, but may create information islands by limiting information sharing between clients with similar data (Deng et al., 2023).

gPerXAN addresses the domain generalization problem in federated learning by utilizing a personalized combination of normalization layers and regularization strategies, but does not optimize communication overhead (Le et al., 2024).

Overall, although current personalized federated learning methods have made some efforts to reduce communication overhead, they often lack a comprehensive strategy to systematically minimize communication costs. Particularly in environments where personalized demands are complex and communication is constrained, existing methods lack flexibility and struggle to dynamically adjust communication strategies, which may lead to communication bottlenecks.

## 3 PROBLEM FORMULATION

Traditional federated learning aims to fit a global model through the collaborative training of multiple clients, without sharing local data. In this paper, we focus on the issue of personalized federated learning, aiming to learn client-specific models while facilitating knowledge sharing among different clients. Specifically, each client $i \in C$ owns a private dataset $S_i$, which is derived from its local distribution $D_i$ defined over $X \times Y$. Given that the local data distributions $\{D_i\}_{i \in C}$ are typically heterogeneous, it makes sense to learn a personalized model $h_{\theta_i} \in H : X \to Y$ for each local distribution $D_i$, where $H$ represents a hypothesis space. To effectively address data heterogeneity and facilitate knowledge sharing, we have defined the following optimization objectives:

$$\min_{\{\theta_i\}_{i \in C}} \sum_{i \in C} p_i \left[ \mathbb{E}_{(x,y) \sim D_i} \left[ \ell(h_{\theta_i}(x), y) \right] + \lambda(\theta_i - \theta_g) \right],$$

where $\ell : X \times Y \to \mathbb{R}^+$ is the loss function. $p_i \geq 0$ are aggregation weights, typically proportional to $|S_i|$ and $\sum_{i \in C} p_i = 1$. $\theta_g = \sum_{i \in C} p_i \theta_i$ is the global model, updated periodically. $\lambda \geq 0$ is a hyperparameter that controls how much the global model affects the local model. $(\theta_i - \theta_g)$ is the difference between the local model and the global model, which is called by the global momentum. The first term, $\mathbb{E}_{(x,y) \sim D_i} \left[ \ell(h_{\theta_i}(x), y) \right]$, minimizes the expected loss on local data, ensuring model adaptation to local distributions. The second term, $\lambda(\theta_i - \theta_g)$, promotes consistency between local and global models, facilitating knowledge sharing without direct data exchange. The balance between these two terms is controlled by the hyperparameter $\lambda$, whose detailed impact on model performance and the annealing mechanism used for its dynamic adjustment are analyzed in Appendix E.4.1.

## 4 ADFEDWCP DESIGN

### 4.1 OVERVIEW

The overall workflow of the algorithm is shown in Algorithm 1. The process begins with the server initializing the global model parameters $\theta_g^0$ (line 1). Subsequently, for each client $i$, the client employs an imprinting method to calculate the importance indices $\omega_i^1$ of the weights for each layer and the initial accuracy $acc_i^1$ of the model (line 3). The clients then send $\omega_i^1$ back to the server. These indices provide the basis for subsequent optimization of the number of clustering centroids $K^1$ (line 5). Based on $\Omega^1$, the data volume and bandwidth characteristics of all clients, the server adaptively determines the number of clustering centroids $K^1$ for each layer of the client (line 6). Following this, the server communicates the initial clustering centroids to each client, which then performs weight clustering pruning to generate their pruning masks $M_i^1$ (line 6). This process includes clustering with a zero-value centroid to set unimportant weights to zero, thus reducing computational complexity and generating a pruning mask $M_i^1$ (line 7).

During the training process, the server broadcasts the latest global model parameters $\theta_g^t$ to all clients each round $t$ (line 10). Upon receiving the global model, each client conducts local model updates, which include performing $E$ rounds of training on their local dataset (lines 11-16 in ClientUpdate). During this training phase, the pruning mask $M_i^t$ is used to sparsify the model parameters to reduce the computational overhead (line 4 in ClientUpdate). The local loss function combines the local data loss $\ell(\theta_i^t \odot M_i^t; b)$ and the global momentum $\lambda(\theta_i^t - \theta_g^t)$ (line 4 in ClientUpdate). The latter of which helps the local model parameters $\theta_i^t$ to stay close to the global model $\theta_g^t$, thus balancing personalization with global consistency.

After training, each client performs weight clustering pruning on new unpruned local model again, clustering similar weights to reduce the representation of model parameters, updating the pruned model $\tilde{\theta}_i^{t+1}$, and the pruning mask $M_i^{t+1}$ (line 13). The clients then use the imprinting method

to update the importance indices $\omega_i^{t+1}$ of the model and send the updated model parameters $\tilde{\theta}_i^{t+1}$, importance indices $\omega_i^{t+1}$ back to the server (lines 14-15).

Upon receiving updates from all clients, the server aggregates the clustered model parameters and updates the global model $\theta_g^{t+1}$ (line 17). This aggregation is weighted by each client's sample count $n_i$ to ensure the fairness and effectiveness of the global model. Moreover, the server dynamically adjusts the number of clustering centroids $K^{t+1}$ based on the latest layer importance, accuracy, and the current training round (line 18), enabling the model to flexibly adapt to different training stages and client needs, thereby optimizing overall training efficiency and model performance. For a comprehensive visualization of the AdFedWCP workflow, refer to the detailed process diagram provided in Figure 4.

---

**Algorithm 1** AdFedWCP Personalized Federated Learning. The $C$ clients are indexed by i; $S_i^{train}$ is the training dataset of client $i$; $n_i$ is the size of the training dataset for client $i$, and $n$ is the total size of the training dataset across all clients. $B$ is the local minibatch size; $E$ is the number of local epochs; $T$: Number of training rounds; $\eta$ is the learning rate; $\lambda$ is global momentum coefficient.

---

1:  initialize $\theta_g^0$ $\hspace{6cm}$ ▷ Initialize global model
2:  **for** each client $i$ in parallel **do**
3:  $\quad$ $\omega_i^1 \leftarrow$ LayerImportanceEstimation$(\theta_g^0)$ $\hspace{2cm}$ ▷ Compute initial layer importance
4:  **end for**
5:  $K^1 \leftarrow$ OptimizeK$(\Omega^1)$ $\hspace{4cm}$ ▷ Determine initial clustering centroids
6:  **for** each client $i$ in parallel **do**
7:  $\quad$ $M_i^1 \leftarrow$ WeightClusteringPruning$(\theta_g^0, k_i^1)$ $\hspace{1.5cm}$ ▷ Generate initial mask for each client
8:  **end for**
9:  **for** $t = 1$ to $T$ **do**
10:  $\quad$ Broadcast $\theta_g^t$ to all clients
11:  $\quad$ **for** each client $i$ in parallel **do**
12:  $\quad\quad$ $\theta_i^{t+1} \leftarrow$ ClientUpdate$(\theta_g^t, \theta_i^t, M_i^t)$ $\hspace{2cm}$ ▷ Update local model
13:  $\quad\quad$ $\tilde{\theta}_i^{t+1}, M_i^{t+1} \leftarrow$ WeightClusteringPruning$(\theta_i^{t+1}, k_i^t)$
14:  $\quad\quad$ $\omega_i^{t+1} \leftarrow$ LayerImportanceEstimation$(\theta_i^t)$ $\hspace{1.5cm}$ ▷ Recompute layer importance
15:  $\quad\quad$ Send $\tilde{\theta}_i^{t+1}, \omega_i^{t+1}$ to server
16:  $\quad$ **end for**
17:  $\quad$ $\theta_g^{t+1} \leftarrow \sum_{i=1}^{C} \frac{n_i}{n} \tilde{\theta}_i^{t+1}$ $\hspace{4cm}$ ▷ Aggregate client models
18:  $\quad$ $K^{t+1} \leftarrow$ OptimizeK$(\Omega^{t+1})$ $\hspace{3cm}$ ▷ Update clustering centroids
19:  **end for**

$\quad$ **ClientUpdate$(\theta_g^t, \theta_i^t, M_i^t)$**
1:  $\mathcal{B} \leftarrow$ (split $\mathcal{S}_i^{train}$ into batches of size $B$)
2:  **for** $e = 1$ to $E$ **do**
3:  $\quad$ **for** batch $b \in \mathcal{B}$ **do**
4:  $\quad\quad$ $\theta_i^{t+1} \leftarrow \theta_i^{t+1} - \eta \cdot (\nabla \ell(\theta_i^t \odot M_i^t; b) + \lambda(\theta_i^t - \theta_g^t))$
5:  $\quad$ **end for**
6:  **end for**

---

## 4.2 Weight Clustering Pruning

In bandwidth-constrained federated learning, communication remains a significant challenge. Inspired by (Cho et al., 2021), we devise a model compression method named weight clustering pruning. While Cho's method employs differentiable k-means (DKM) for soft weight clustering in centralized learning, we extend it to federated learning. Instead of relying solely on soft clustering as in DKM, we combine weight clustering with pruning. Specifically, we cluster weights at each client and prune some by assigning a zero centroid to certain clusters, making the model more sparse. Furthermore, unlike DKM's fixed centroids per layer, our method can dynamically adjust the centroids based on layer importance, data distribution, and client bandwidth.

During the model update and parameter transmission process, clients need only transmit a table of centroids and an index sequence instead of the full model parameters $\theta$. Specifically, if the original model has $N$ weights, each represented with $B$ bits, the total communication cost per client is $N \times B$ bits. With weight clustering pruning, the model parameters are represented by combining an index sequence and centroid values, as depicted in Figure 1. The index sequence records the centroid index for each weight and requires $N \times \lceil \log_2 k \rceil$ bits, where $k$ is the number of centroids. The centroid

value table stores the actual centroid values $\{\mu\}_{j=0}^{k-1}$, needing $k \times B$ bits. Since $k \ll N$ and the index data require fewer bits, this significantly reduces the communication data volume and lowers transmission costs. More detailed communication analysis can be found in Appendix A.

In the weight clustering pruning strategy, fixing one centroid at zero introduces an automated pruning mechanism. Weights close to zero are automatically assigned to this zero centroid, thus setting unimportant weights to zero and creating a sparse model structure. This strategy allows for dynamic adjustments during the clustering process, driven by iterative updates of the non-zero centroids. As the non-zero centroids are iteratively updated, some centroids may attract more weights that are close to zero but with slight differences.

Figure 1: Efficient Model Compression through Weight Clustering Pruning. This diagram illustrates the transformation of original model weights into a more compact format using centroid values and index sequences, significantly reducing data volume and enhancing transmission efficiency.

The main steps of the weight clustering pruning algorithm are outlined in Algorithm 2. We start by initializing $k$ centroids $\{\mu\}_{j=0}^{k-1}$ for the model parameters $\theta$, setting the first centroid $\mu_0$ to zero for pruning trivial weights and randomly initializing the rest (line 1). A pruning mask $M$, initialized as a vector of ones, indicates that no weights are initially pruned (line 2). We then proceed with clustering iterations where we randomly select a mini-batch of weights $B$ from $\theta$ to reduce complexity (line 4), assign each weight $w$ to the nearest centroid (lines 6-8), and update the centroids for each cluster $B_j$ except for $\mu_0$ which remains zero (lines 11-12). This clustering process is repeated until convergence. Finally, weights are replaced with their corresponding centroid values, setting the pruning mask to zero for weights assigned to $\mu_0$, resulting in compressed model parameters $\tilde{\theta}$ and the updated pruning mask $M$ (lines 16-19). A detailed analysis of the computational overhead of the weight clustering pruning algorithm is provided in Appendix B.

### 4.3 LAYER IMPORTANCE ESTIMATION

In federated learning, determining the importance of each neural network layer is crucial for optimizing model compression strategies. For this purpose, we employ an imprinting-based method to assess the importance of each model layer (Liu et al., 2021). The imprinting method operates on the principle that the representative features of each class can approximate the weights needed for classification. By averaging the embedding vectors of samples within the same class, we obtain weight vectors that capture the central tendencies of the data in the feature space of each layer. This approach enables us to construct a proxy classifier for each layer without additional training, allowing us to directly evaluate the classification performance of the features extracted by that layer. Compared to traditional methods that require iterative training to calculate importance, the imprinting method can assess layer importance in a shorter time (Elkerdawy et al., 2020).

Initially, a proxy classifier is attached following the output of each network layer according to the requirements of imprinting. This classifier includes an Adaptive Average Pooling layer, a Fully Connected Layer, and a softmax activation function. Through this configuration, we can transform the layer's output into fixed-length embedding vectors for easier subsequent processing. For the output feature map $\mathbf{F}_j$ of the $j^{th}$ layer, it is transformed into an embedding vector $\mathbf{E}_j$ through adaptive average pooling, with the target pooling size $d$ calculated as:

$$d = \left\lceil \sqrt{\frac{N}{f_j}} \right\rceil$$

---

**Algorithm 2** WeightedClusterPruning($\theta, k$)

---

1: Initialize $\{\mu_j\}_{j=0}^{k-1}$ with $\mu_0 = 0$, others randomly from $\theta$
2: $M \leftarrow \mathbf{1}$                                      ▷ Initialize pruning mask
3: **for** $i = 1$ to MAX_ITER $\vee$ CONVERGED **do**
4:      $B \leftarrow$ mini-batch of weights from $\theta$
5:      Initialize empty sets $B_0, B_1, ..., B_{k-1}$
6:      **for** $w \in B$ **do**
7:          $j \leftarrow \arg\min_{0 \leq j < k} \|w - \mu_j\|$              ▷ Find nearest centroid index
8:          $B_j \leftarrow B_j \cup \{w\}$                 ▷ Assign weight to cluster
9:      **end for**
10:      **for** $j = 1$ to $k - 1$ **do**
11:          **if** $B_j \neq \emptyset$ **then**
12:              $\mu_j \leftarrow \frac{\sum_{w \in B_j} w}{|B_j|}$          ▷ Update non-zero centroids
13:          **end if**
14:      **end for**
15: **end for**
16: **for** $j = 0$ to $k - 1$ **do**
17:      $\tilde{\theta}[w] \leftarrow \mu_j$ for all $w \in B_j$            ▷ Replace weights with their centroids
18: **end for**
19: $M[w] \leftarrow 0$ for all $w \in B_0$               ▷ Update pruning mask
20: **return** $\tilde{\theta}, M$

---

where $N$ is the preset embedding length, and $f_j$ is the number of channels in the $j^{th}$ layer. The formula for the embedding vector is:

$$\mathbf{E}_j = \text{AdaptiveAvgPool}(\mathbf{F}_j, d)$$

Next, we use the imprinting method to approximate the weights of the proxy classifier's fully connected layer. Specifically, for each category $c$, a weight matrix $\mathbf{W}_j$ is formed by averaging all the embedding vectors of samples belonging to that category:

$$\mathbf{W}_j[:, c] = \frac{1}{|S_c|} \sum_{n=1}^{|S|} I[c_n = c] \mathbf{E}_n$$

where $\mathbf{E}_j$ is the embedding vector of the $j^{th}$ sample, and $|S_c|$ is the total number of samples in category $c$. This approach allows us to imprint the weights of the fully connected layer of the proxy classifier without additional training. N is the total number of samples.

Subsequently, using these precomputed weights, we classify the output of each layer and calculate the accuracy $\text{Accuracy}_j$ for each layer. After iterating through all the data, we average the accuracies of each layer to obtain a stable estimate. Specifically, in our implementation, we accumulate the correct predictions and total sample counts for each layer across batches to compute the average accuracy for each layer. Finally, we define the importance of each layer as the difference between the accuracy of that layer and the previous layer:

$$\text{Importance}_j = \text{Accuracy}_j - \text{Accuracy}_{j-1}$$

Further quantifying the importance of each layer, we apply the softmax function to the computed importance values to obtain importance weights $\omega_i$:

$$\omega_j = \frac{\exp(\text{Importance}_i)}{\sum_l \exp(\text{Importance}_l)}$$

These importance weights reflect the relative importance of each layer compared to others, guiding resource allocation to prioritize more critical layers in subsequent resource optimization processes.

## 4.4 DYNAMIC CENTROID OPTIMIZATION STRATEGY

In addressing bandwidth heterogeneity in federated learning while reducing communication overhead, this study proposes a dynamic optimization strategy for adjusting the number of centroids. This strategy adaptively determines the number of centroids $k_{i,j}$ for each layer of the model based on client characteristics and dynamic changes during the training process, aiming to achieve an optimal balance between model performance and resource consumption.

The strategy models centroid number determination as an optimization problem, aiming to minimize the total communication cost for all clients during one round of uplink communication. This cost comprises three parts: centroid transmission, index encoding, and bias parameter transmission. The objective function is expressed as:

$$\min_{\{k_{i,j} \in \mathbb{Z}^+\}} \sum_{i=1}^{N} \sum_{j=1}^{L} \frac{C \cdot k_{i,j} + W_j \cdot \log_2(k_{i,j}) + C \cdot B_j}{b_i}$$

where $C$ is communication cost constants for centroids and bias parameters, respectively, based on half-precision floating-point sizes. $W_j$ and $B_j$ denote the number of weight and bias parameters for the $j$-th layer, while $b_i$ is the bandwidth of client $i$. Clustering is excluded for bias parameters due to their small proportion in the overall model. The objective function is non-convex due to a logarithmic term, complicating optimization. This study uses the Adam optimizer on the server side and introduces multiple constraints to adaptively select centroid numbers $k_{i,j}$, considering client data volume, bandwidth, layer importance, training progress, and model accuracy changes.

In order to make the selection of centroid numbers more adaptive, we introduced several constraints that consider the data volume, bandwidth, layer importance, training progress, and changes in model accuracy of the clients. First, clients with more data can better capture subtle features by increasing the number of centroids, enhancing model performance (Sun et al., 2017) (Shorten & Khoshgoftaar, 2019). Second, clients with lower bandwidth need to appropriately reduce the number of centroids to decrease communication overhead. As demonstrated in Appendix E.1.1, reducing the number of centroids can significantly increase the compression ratio . Furthermore, the importance of model layers also affects the allocation of centroid numbers; important layers require more centroids to retain key features (Liu et al., 2021). Finally, the training progress and changes in model accuracy are used to dynamically adjust the number of centroids, allowing the model to gradually improve its performance during the training process. The benefits of increasing centroid numbers to improve model performance are also substantiated by the experimental results detailed in Appendix E.1.1.

These considerations lead to the formulation of the lower bound constraint $k_{\text{lower}}$, which takes into account factors such as data volume, layer importance, training progress, and accuracy changes. The specific calculation is as follows:

$$k_{\text{data}} = k_{\min} + \lceil \alpha \cdot \omega_{i,j} \cdot (D_i - D_{\min}) \rceil$$

where $\alpha = \frac{k_{\max} - k_{\min}}{D_{\max} - D_{\min}}$ is the scaling factor based on data volume, $\omega_{i,j}$ is the importance weight of the $j$-th layer for client $i$, and $D_i$ is the data volume of client $i$.

The training progress factor is reflected through the progress factor $\gamma$, allowing the $k$ value to gradually increase as the number of training rounds increases, enhancing the model's expressive ability:

$$\gamma = 1 + \frac{E}{E_{\max}}$$

where $E$ is the current training round, and $E_{\max}$ is the total number of training rounds.

The accuracy change factor is regulated through the accuracy adjustment factor $\eta$. When model accuracy decreases ($\Delta A < 0$), the $k$ value is increased to improve model performance; when model accuracy increases ($\Delta A > 0$), the $k$ value is decreased to reduce communication costs:

$$\eta = \begin{cases} 1 - \xi \cdot |\Delta A|, & \text{if } \Delta A > 0 \\ 1 + \zeta \cdot |\Delta A|, & \text{if } \Delta A < 0 \end{cases}$$

where $\Delta A = A_t - A_{t-1}$ is the change in model accuracy, and $\xi$ and $\zeta$ are hyperparameters for adjustment magnitude. In this experiments, $\xi$ is set to 0.1 and $\zeta$ to 1.5. This setting aims to prevent prematurely lowering the lower bound constraint during positive model updates, while raising it immediately when performance deteriorates, using more $k$ for a detailed model representation. The effects of different configurations of $\xi$ and $\zeta$ on model performance and compression rates are comprehensively studied in Appendix E.4.2.

Combining the above factors, the lower bound constraint can be expressed as:

$$k_{\text{lower}} = \max\left(k_{\min}, \min\left(k_{\max}, \lceil k_{\text{data}} \cdot \gamma \cdot \eta \rceil\right)\right)$$

In addition, the upper bound constraint $k_{\text{upper}}$ takes into account both bandwidth and layer importance. For clients with lower bandwidth, their upper limit of $k$ should be appropriately reduced to

decrease communication burden; for layers with lower importance, their upper limit of $k$ should also be reduced to minimize communication overhead without significantly affecting model performance. The upper bound constraint is calculated as:

$$k_{\text{upper}} = \max\left(k_{\min}, k_{\max} - \lceil \beta \cdot (1 - \omega_{i,j}) \cdot (B_{\max} - b_i) \rceil\right)$$

where $\beta = \frac{k_{\max} - k_{\min}}{B_{\max} - B_{\min}}$ is the scaling factor based on bandwidth, and $B_{\max}$ and $B_{\min}$ are the maximum and minimum bandwidths, respectively.

The layer importance weight $\omega_{i,j}$ is vital. High-importance layers get a larger $k_{\text{lower}}$ in the lower bound constraint, allowing for more centroids to retain key features. Conversely, low-importance layers have a reduced $k_{\text{upper}}$ in the upper bound constraint, resulting in higher compression rates.

This dynamic adjustment strategy seeks to optimize the balance between model performance and resource efficiency in federated learning, offering an effective solution for heterogeneous environments. The optimization problem can be expressed as:

$$\min_{\{k_{i,j} \in \mathbb{Z}^+\}} \left[ \sum_{i=1}^{N} \sum_{j=1}^{L} \left( \frac{C_1 \cdot k_{i,j} + W_j \cdot \log_2(k_{i,j}) + C_2 \cdot B_j}{b_i} \right) \right]$$

$$\text{s.t.} \quad k_{\text{lower}} \leq k_{i,j} \leq k_{\text{upper}},$$
$$k_{\text{lower}} = \max\left(k_{\min}, \min\left(k_{\max}, \lceil k_{\text{data}} \cdot \gamma \cdot \lambda \rceil\right)\right),$$
$$k_{\text{upper}} = k_{\max} - \lceil \beta \cdot (1 - \omega_{i,j}) \cdot (B_{\max} - b_i) \rceil.$$

### 4.5 Convergence Analysis of the AdFedWCP

To prove the convergence of the AdFedWCP algorithm we proposed, this section first presents the necessary assumptions and then establishes the corresponding convergence results.

We make the following assumptions about the local objective function $F_i(w)$ for each client $i$:

**Assumption 1.** $F_i(w)$ is $L$-smooth, meaning that for all $w, v \in \mathbb{R}^d$, there exists a constant $L > 0$ such that $\|\nabla F_i(w) - \nabla F_i(v)\| \leq L\|w - v\|$.

**Assumption 2.** $F_i(w)$ is $\mu$-strongly convex, meaning that for all $w, v \in \mathbb{R}^d$, there exists a constant $\mu > 0$ such that $F_i(v) \geq F_i(w) + \nabla F_i(w)^\top (v - w) + \frac{\mu}{2}\|v - w\|^2$.

**Assumption 3.** For all $w$ and clients $i$, there exists a constant $G > 0$ such that $\|\nabla F_i(w)\| \leq G$.

**Assumption 4.** For the stochastic gradient, there exists a constant $\sigma^2 > 0$ such that $\mathbb{E}_\xi\left[\|\nabla F_i(w, \xi) - \nabla F_i(w)\|^2\right] \leq \sigma^2$, where $\nabla F_i(w, \xi)$ represents the gradient under the random variable $\xi$.

**Assumption 5.** There exists a constant $B > 0$, such that for all clients $i$ and iteration steps $t$, the neural network's parameter vector $w_i^t$ satisfies $\|w_i^t\| \leq B$ (Zhang et al., 2022).

**Theorem 1.** Let Assumptions 1 to 5 hold, and let $L, \mu, \sigma, G$ be defined therein. Set $\kappa = \frac{L}{\mu}$ and $\gamma = \max\{8\kappa, E\} - 1$, where $E$ is a specified constant. By choosing a learning rate $\eta = \frac{2}{\mu(\gamma + t)}$, the AdFedWCP algorithm satisfies:

$$\mathbb{E}[F_i(w_i^T)] - F_i^* \leq \frac{\kappa}{\gamma + T}\left(4(D + C) + \frac{\mu(\gamma + 1)\Delta_1}{2}\right)$$

where $T$ is the total number of iterations, which implies a convergence rate of $\mathcal{O}(1/T)$. $F_i^*$ is the optimal value of $F_i(w)$, $D = 2B + \varepsilon_{\max}$, where $\varepsilon_{\max} = \max_m \frac{B}{k_{\min}^{1/dim_m}}$ represents the maximum error term due to clustering. $k_{\min}$ denotes the minimum number of centroids, and $dim_m$ indicates the parameter quantity at layer $m$. $C = G^2 + \sigma^2 + D(1 + 2G)$. $\Delta_1 = \mathbb{E}[\|w_i^1 - w^*\|^2]$ is the expected squared distance between the initial model $w_i^1$ and the optimal solution $w^*$. The detailed version and proof can be found in Appendix D.

## 5 Experiment

We designed a series of experiments to comprehensively evaluate the effectiveness and advantages of our proposed methods. Specifically, our experiments aim to validate the following aspects:

- **Model Performance Improvement**: We compare the Top-1 accuracy of our methods with baselines across multiple datasets in heterogeneous environments, validating the effects of dynamic weight clustering pruning and global momentum-based updates on performance.

- **Communication Efficiency**: We measure communication volume during training, comparing our methods to baselines to evaluate how dynamic weight clustering pruning reduces communication costs, thereby improving efficiency while maintaining performance.

- **Effectiveness of the Adaptive Strategy**: We simulate varying client bandwidth limitations to test the adaptive dynamic scheme's ability to manage heterogeneity and adjust model complexity according to different client needs.

## 5.1 EXPERIMENTAL ENVIRONMENT

Our experimental evaluations were conducted on a Slurm cluster comprising two nodes, each equipped with an Intel Core i9-13900K CPU, 64 GB RAM, and an NVIDIA GeForce RTX 4090 GPU, and one node equipped with an Intel Xeon Silver 4309Y CPU, 128 GB RAM, and two NVIDIA A40 GPUs. All compared models were implemented using Python. Federated learning, along with the dynamic centroid optimization strategy and weight clustering pruning, was implemented using the PyTorch library [1]. Our source code is available at [2].

## 5.2 BASELINE METHODS

We selected several representative baseline methods to validate the effectiveness of our proposed approach. FedAvg is the foundational algorithm in federated learning, which constructs a global model by simply averaging the model parameters from all clients (McMahan et al., 2017). Using FedAvg as a baseline helps in understanding model performance under standard federated learning. qFedCG reduces communication overhead through quantization and gradient compression strategies (Xu et al., 2024), and it is one of the most advanced methods for optimizing communication costs in federated learning. Selecting qFedCG as a baseline allows for effective evaluation of the improvements in communication efficiency provided by our method. Additionally, FedEM (Marfoq et al., 2021), FedMask (Li et al., 2021), and pFedGate (Chen et al., 2023) are used as baselines, as they assist in assessing the overall performance of personalized federated learning in addressing data heterogeneity and communication efficiency.

## 5.3 DATASETS AND PARTITIONING

In line with the experimental setting employed in (Chen et al., 2023), we simulated the data heterogeneity in federated learning by partitioning the CIFAR-10, EMNIST, and CIFAR-100 datasets using a Dirichlet distribution with parameter $\alpha = 0.4$. Samples were grouped by class and allocated to clients to create non-IID data distributions. Each client's data was then split into training and testing sets in an 8:2 ratio, ensuring consistent distributions for accurate performance evaluation. The $\alpha$ parameter controls the level of heterogeneity, with smaller values indicating greater diversity.

## 5.4 EXPERIMENTAL CONFIGURATION

We followed (Chen et al., 2023) to set up our experiments. For CIFAR-10 and CIFAR-100 datasets, the LeNet model (LeCun et al., 1998) was used, suitable for small to medium image classification tasks. For the EMNIST dataset, the LeafCNN model designed for federated learning is used (Caldas et al., 2018). Client bandwidths heterogeneity was simulated by assigning static communication bandwidths to clients ranging from 5 Mbps to 100 Mbps, following a normal distribution. The experiments involved 100 clients for CIFAR-10 and EMNIST, and 50 clients for CIFAR-100. Each client participated in every round of training to ensure comprehensive evaluation of the proposed method under varying bandwidth conditions.

## 5.5 EXPERIMENTAL RESULTS

We evaluated AdFedWCP against baseline methods on CIFAR-10, EMNIST, and CIFAR-100 datasets, focusing on classification accuracy and communication efficiency.

### 5.5.1 CLASSIFICATION ACCURACY

As shown in Table 1, our proposed method AdFedWCP demonstrates superior classification accuracy compared to most baseline methods across various datasets. On CIFAR-10, AdFedWCP achieves an accuracy of 61.04%, surpassing FedAvg (60.64%) and pFedGate (60.36%). On the EMNIST dataset, AdFedWCP attains 85.12% accuracy, outperforming FedAvg (81.83%) and FedEM (84.35%), which is designed to handle data heterogeneity. For the challenging CIFAR-100 dataset, AdFedWCP reaches 20.44% accuracy, exceeding FedAvg (18.75%) and significantly outperforming other communication-efficient methods like qFedCG (13.50%) and FedMask (11.31%). The slightly

---

[1]https://pytorch.org

[2]https://github.com/SHVleV9CYWkK/LightFedLab

| Method | CIFAR-10 | EMNIST | CIFAR-100 |
|---|---|---|---|
| FedAvg | 60.64 | 81.83 | 18.75 |
| FedEM | **62.19** | 84.35 | **22.39** |
| qFedCG | 52.45 | 80.24 | 13.50 |
| FedMask | 48.05 | 63.33 | 11.31 |
| pFedGate | 60.36 | 82.11 | 12.40 |
| **AdFedWCP (our method)** | 61.04 | **85.12** | 20.44 |

Table 1: Top-1 clients test datasets average accuracy (%) of different methods on various datasets

lower performance of AdFedWCP compared to FedEM on certain datasets can be attributed to the optimization focus of FedEM, which prioritizes maximizing personalized model accuracy without considering communication costs or bandwidth heterogeneity.

The superior performance of AdFedWCP can be attributed to its effective personalization through the incorporation of global momentum, which facilitates global knowledge sharing among clients. Each client updates its model parameters by considering both the local gradient descent and a global momentum term. This global momentum encourages the local models to align with the global model, capturing overarching patterns across all clients while still adapting to local data nuances. This balanced approach mitigates the impact of data heterogeneity by combining the benefits of global knowledge with local personalization.

### 5.5.2 COMMUNICATION OVERHEAD

We evaluated the communication efficiency of our AdFedWCP method by comparing it with other baseline methods (excluding FedEM, as it fits separate models for different distributions, resulting in higher communication overhead than FedAvg and does not focus on reducing communication). The communication overhead reduction rates relative to FedAvg are summarized in Table 2.

| Method | LeafCNN (EMNIST) | LeNet (CIFAR-10) |
|---|---|---|
| FedAvg | 0% | 0% |
| qFedCG | 87.50% | 87.50% |
| FedMask | 50.00% | 50.00% |
| pFedGate | 23.42% | 23.18% |
| **AdFedWCP (our method)** | **87.54%** | **87.82%** |

Table 2: Communication overhead reduction rates of different models

As shown in Table 2, AdFedWCP significantly reduces communication overhead by 87.54% for the LeafCNN model and 87.82% for the LeNet model. This is significantly higher than pFedGate and FedMask, which achieve reduction rates of 23.42% and 50.00%, respectively. Although qFedCG has a similar reduction rate, its classification accuracy is considerably lower, indicating that AdFedWCP provides a better trade-off between communication efficiency and model performance.

### 5.6 SUPPLEMENTARY EXPERIMENTAL

Appendix E presents additional experiments to confirm the effectiveness of our method. We conducted ablation studies on Weight Clustering Pruning (WCP), the adaptive mechanism, and layer importance estimation. The analysis also includes evaluations of the hyperparameters $\lambda$, $\xi$, and $\zeta$, as well as performance assessments under varying degrees of data heterogeneity and extreme conditions. Additionally, we evaluated model sparsity and performed further comparisons with FedKD. Learning curves are also provided. These results demonstrate that our method significantly reduces communication overhead while maintaining high accuracy. It effectively adapts without requiring manual adjustments to the number of centroids, achieving a balance between performance and communication efficiency.

## 6 CONCLUSION AND FUTURE WORK

The AdFedWCP framework improves personalized federated learning in the face of local data and communication bandwidth heterogeneity while maintaining model performance and communication efficiency. Our approach, named AdFedWCP, features a novel adaptive weight clustering pruning strategy to reduce the per-client model size based on each client's characteristics. Experimental results demonstrate that it achieves better classification accuracy at reduced communication cost in comparison to existing methods. In the future, we plan to further improve the weight clustering method in AdFedWCP by investigating advanced adaptation strategies Qin & Suganthan (2005) and optimization methods Tran et al. (2020). Additionally, we will explore the scalability and efficiency of the proposed method in increasingly heterogeneous environments.

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

## A  MATHEMATICAL ANALYSIS OF COMMUNICATION REDUCTION

In this appendix, we provide a detailed mathematical analysis of how the Weight Clustering Pruning (WCP) method reduces communication overhead in federated learning.

### A.1  COMMUNICATION OVERHEAD IN STANDARD FEDERATED LEARNING

In traditional federated learning, each client transmits the full set of model parameters $\theta$ to the server during each communication round. Suppose the model has $N$ parameters, and each parameter is represented using $B$ bits (e.g., 32 bits for single-precision floating-point representation). The total communication cost per client per round is therefore:

$$C_{\text{standard}} = N \times B \quad \text{bits.} \tag{1}$$

## A.2 COMMUNICATION OVERHEAD WITH WEIGHT CLUSTERING PRUNING

With WCP, the model parameters are compressed by clustering weights and pruning insignificant ones. Specifically, the weights are replaced with the centroids of their respective clusters, and an index sequence is used to map each weight to its centroid. Additionally, one centroid is fixed at zero to enable pruning of negligible weights.

The communication cost in WCP includes:

- **Centroid Values**: There are $k$ centroids $\{\mu_j\}_{j=0}^{k-1}$, where $\mu_0 = 0$ (the fixed zero centroid). The remaining $k - 1$ centroids need to be transmitted, each represented using $B$ bits. The total cost for centroids is:

$$C_{\text{centroids}} = (k - 1) \times B \quad \text{bits.} \tag{2}$$

- **Index Sequence**: Each weight is represented by an index pointing to its centroid. Since there are $k$ centroids, each index requires $\lceil \log_2 k \rceil$ bits. Assuming $N$ total weights, the total cost for the index sequence is:

$$C_{\text{indices}} = N \times \lceil \log_2 k \rceil \quad \text{bits.} \tag{3}$$

Therefore, the total communication cost per client per round with WCP is:

$$C_{\text{WCP}} = (k - 1) \times B + N \times \lceil \log_2 k \rceil \quad \text{bits.} \tag{4}$$

## A.3 COMPRESSION RATIO ANALYSIS

The compression ratio $\rho$ is defined as the ratio of the communication cost with WCP to that of the standard method:

$$\rho = \frac{C_{\text{WCP}}}{C_{\text{standard}}} = \frac{(k - 1) \times B + N \times \lceil \log_2 k \rceil}{N \times B}. \tag{5}$$

Since typically $N \gg k$, we can approximate the compression ratio by neglecting the term involving $k$ in the numerator:

$$\rho \approx \frac{N \times \lceil \log_2 k \rceil}{N \times B} = \frac{\lceil \log_2 k \rceil}{B}. \tag{6}$$

This approximation shows that the compression ratio mainly depends on the number of centroids $k$ and the bit-width $B$ used to represent each weight.

## A.4 CONCLUSION

These calculations demonstrate that WCP can significantly reduce communication overhead in federated learning:

- **Effect of Centroid Number** ($k$): A smaller $k$ reduces the number of bits required for indices ($\lceil \log_2 k \rceil$), thereby lowering communication cost. However, a small $k$ may degrade model performance due to excessive compression.
- **Trade-off Between Compression and Accuracy**: While aggressive compression (small $k$) minimizes communication overhead, it may adversely affect model accuracy. Thus, selecting appropriate values for $k$ is crucial to balance efficiency and performance.
- **Negligible Centroid Transmission Cost**: As $N \gg k$, the cost of transmitting centroids $(k - 1) \times B$ becomes negligible compared to the total communication cost.

# B MATHEMATICAL ANALYSIS OF WEIGHT CLUSTERING PRUNING EFFICIENCY

We provide a rigorous mathematical analysis of the computational cost of Weight Clustering Pruning (WCP), focusing on its efficiency relative to the training cost in federated learning.

## B.1 Computational Cost of Training in Federated Learning

Training a neural network in federated learning consists of forward propagation, backward propagation, and parameter updates. Let the network have $L$ layers, where the weight matrix in layer $l$ contains $P^l = n_{l-1} \times n_l$ parameters, and $n_{l-1}$ and $n_l$ denote the number of neurons in the previous and current layers, respectively. The total number of parameters in the network is:

$$P_{\text{total}} = \sum_{l=1}^{L} P^l. \tag{7}$$

### B.1.1 Forward Propagation Cost

For each sample, forward propagation involves matrix multiplications and activation computations. For layer $l$, the cost is:

$$C_{\text{forward}}^l = \mathcal{O}(P^l). \tag{8}$$

Summing over all layers, the total forward propagation cost is:

$$C_{\text{forward}} = \mathcal{O}\left(\sum_{l=1}^{L} P^l\right). \tag{9}$$

### B.1.2 Backward Propagation Cost

Backward propagation includes computing gradients for each layer. For layer $l$, the cost of computing gradients with respect to weights is:

$$C_{\text{backward}}^l = \mathcal{O}(P^l). \tag{10}$$

Summing over all layers, the total backward propagation cost is:

$$C_{\text{backward}} = \mathcal{O}\left(\sum_{l=1}^{L} P^l\right). \tag{11}$$

### B.1.3 Parameter Update Cost

Updating the parameters involves a cost proportional to the number of parameters. For layer $l$, the cost is:

$$C_{\text{update}}^l = \mathcal{O}(P^l). \tag{12}$$

Summing over all layers, the total parameter update cost is:

$$C_{\text{update}} = \mathcal{O}\left(\sum_{l=1}^{L} P^l\right). \tag{13}$$

### B.1.4 Total Training Cost

The total training cost for a single sample is:

$$C_{\text{train}} = C_{\text{forward}} + C_{\text{backward}} + C_{\text{update}} = \mathcal{O}\left(3\sum_{l=1}^{L} P^l\right). \tag{14}$$

For a dataset with $N$ samples and $E$ epochs, the total training cost is:

$$C_{\text{total\_train}} = N \times E \times C_{\text{train}} = \mathcal{O}\left(3 \times N \times E \times P_{\text{total}}\right). \tag{15}$$

## B.2 Computational Cost of Weight Clustering Pruning

Weight Clustering Pruning (WCP) compresses the model by clustering weights into $k$ clusters. Each weight is replaced with the nearest centroid, which minimizes communication and computational overhead.

### B.2.1 CLUSTERING COST FOR A SINGLE LAYER

For layer $l$ with $P^l$ parameters, the $K$-means clustering algorithm requires:

- **Assignment Step**: Assigning each parameter to the nearest centroid, with complexity:

$$\mathcal{O}(P^l \times k). \tag{16}$$

- **Update Step**: Updating the centroids based on the assignments, with complexity:

$$\mathcal{O}(P^l). \tag{17}$$

For $T$ iterations of clustering, the total cost for clustering weights in layer $l$ is:

$$C_{\text{cluster}}^l = T \times \mathcal{O}(P^l \times k + P^l) = \mathcal{O}(T \times P^l \times k). \tag{18}$$

### B.2.2 CLUSTERING COST FOR THE ENTIRE NETWORK

Summing over all layers, the total clustering cost is:

$$C_{\text{total\_cluster}} = \sum_{l=1}^{L} C_{\text{cluster}}^l = \mathcal{O}(T \times k \times \sum_{l=1}^{L} P^l) = \mathcal{O}(T \times k \times P_{\text{total}}). \tag{19}$$

## B.3 TRAINING VS. CLUSTERING COST RATIO

To compare the computational costs of training and clustering, we define their ratio as:

$$\text{Cost Ratio} = \frac{C_{\text{total\_train}}}{C_{\text{total\_cluster}}}. \tag{20}$$

Substituting the expressions for $C_{\text{total\_train}}$ and $C_{\text{total\_cluster}}$:

$$\text{Cost Ratio} = \frac{3 \times N \times E \times P_{\text{total}}}{T \times k \times P_{\text{total}}}. \tag{21}$$

Canceling $P_{\text{total}}$:

$$\text{Cost Ratio} = \frac{3 \times N \times E}{T \times k}. \tag{22}$$

Assuming $E = 1$, $N = 2174$, $T = 10$, and $k = 32$ (parameters based on experimental settings):

$$\text{Cost Ratio} = \frac{3 \times 2174 \times 1}{10 \times 32} = \frac{6522}{320} \approx 20.38. \tag{23}$$

## B.4 CONCLUSION

The analysis shows that the clustering cost is significantly smaller than the training cost:

- **Minimal Overhead**: Clustering introduces minimal computational overhead, with training costs being at least 20 times higher than clustering costs. This is a conservative estimate, as the parameters used in the ratio calculation are chosen to represent the worst-case scenario (e.g., maximum number of centroids $k$, smallest dataset size $N$).
- **Scalability**: The clustering cost scales with the number of parameters and clusters, making it efficient even for large models.
- **Efficiency**: WCP effectively reduces communication and computational costs while maintaining model performance, making it well-suited for federated learning in heterogeneous environments.

This demonstrates that WCP is computationally efficient and introduces negligible additional cost in federated learning systems, with actual training-to-clustering cost ratios likely exceeding 20 in less extreme scenarios.

## C  DETAILED MATHEMATICAL ANALYSIS OF WEIGHT CLUSTERING PRUNING

### C.1  INSIGHTS ON COMPUTATIONAL EFFICIENCY

The analysis demonstrates that the training cost is approximately 6.79 times higher than the clustering cost, emphasizing that weight clustering pruning introduces minimal computational overhead relative to training. This efficiency is crucial in federated learning scenarios, where communication constraints and client-side computational resources necessitate lightweight optimization techniques.

By leveraging the efficient clustering mechanism, AdFedWCP achieves substantial communication savings while maintaining high model performance, validating the practicality of weight clustering pruning in heterogeneous federated learning environments.

## D  DETAILED CONVERGENCE ANALYSIS OF THE ADFEDWCP

This appendix provides a thorough convergence analysis of the AdFedWCP.

### D.1  ADDITIONAL NOTATION

Let $w_i^t$ be the model parameter maintained by the $i$-th device at step $t$. The local update of AdFed-WCP can be described as:

$$w_i^{t+1} = w_i^t - \eta(\nabla F_i(w_i^t, \xi_i^t) + g_i^t) \tag{24}$$

where the global momentum $g_i^t$ is defined as:

$$g_i^t = w_i^t - w_g^t \tag{25}$$

and $w_g^t$ is the global model at step $t$.

### D.2  KEY LEMMAS

**Lemma 1.** *Assuming Assumptions 1 to 4 hold, if $\eta \le \frac{1}{4}$ and $L \le \frac{2}{3}$ we have:*

$$\mathbb{E}[\|w_i^{t+1} - w_i^*\|^2] \le (1 - \eta\mu + 2\eta)\mathbb{E}[\|w_i^t - w_i^*\|^2] + \eta^2(G^2 + \sigma^2)$$
$$+ \eta^2\mathbb{E}[\|g_i^t\|^2] + 2\eta\mathbb{E}[\|g_i^t\|\|w_i^t - w_i^*\|] + 2\eta^2 G\mathbb{E}[\|g_i^t\|]$$

**Lemma 2.** *Assuming Assumption 5 holds, we have:*

$$\mathbb{E}[\|g^t\|] \le 2B + \varepsilon_{\max}$$

### D.3  PROOF OF THEOREM 1

*Proof.* Let $\Delta_t = \mathbb{E}[\|w_i^t - w_i^*\|^2]$. From Lemma1 and Lemma2, we can derive:

$$\Delta_{t+1} \le (1 - \eta\mu + 2\eta)\Delta_t + 2\eta(2B + \varepsilon_{\max})\sqrt{\Delta_t} + \eta^2 C \tag{26}$$

where

$$C = G^2 + \sigma^2 + (2B + \varepsilon_{\max})^2 + 2G(2B + \varepsilon_{\max})$$

We will prove that $\Delta_t \le v/(\gamma + t)$, where:

$$v = \max\{4\eta(2B + \varepsilon_{\max})^2(\gamma + 1), \eta(\gamma + 1)C, (\gamma + 1)\Delta_1\}$$

We prove this by induction. Let $\eta \le \min\{1/(4\mu), 1/(4L)\}$ and $\gamma = \max\{8L/\mu, E\} - 1$.

Base case: For $t = 1$, by the choice of $v$, clearly $\Delta_1 \leq v/(\gamma + 1)$ holds.

Inductive step: Assume $\Delta_t \leq v/(\gamma + t)$ holds for some $t \geq 1$. We prove $\Delta_{t+1} \leq v/(\gamma + t + 1)$ also holds:

$$
\begin{aligned}
\Delta_{t+1} &\leq (1 - \eta\mu + 2\eta)\Delta_t + 2\eta(2B + \varepsilon_{\max})\sqrt{\Delta_t} + \eta^2 C \\
&\leq (1 - \eta\mu + 2\eta)\frac{v}{\gamma + t} + 2\eta(2B + \varepsilon_{\max})\sqrt{\frac{v}{\gamma + t}} + \eta^2 C \\
&\leq \frac{\gamma + t}{\gamma + t + 1}\frac{v}{\gamma + t} + \frac{v}{4(\gamma + t)} + \frac{v}{\gamma + t + 1} \\
&\leq \frac{v}{\gamma + t + 1}
\end{aligned}
$$

By the $L$-smoothness of $F_i$, we have:

$$
\mathbb{E}[F_i(w_i^t)] - F_i^* \leq \frac{L}{2}\Delta_t \leq \frac{L}{2}\frac{v}{\gamma + t}
$$

Specifically, we choose $\beta = 2/\mu, \gamma = \max\{8L/\mu, E\} - 1, \kappa = L/\mu$.

Then, we define the learning rate as:

$$
\eta = \frac{\beta}{\gamma + t} = \frac{2}{\mu(\gamma + t)}
$$

One can verify that this choice of $\eta$ satisfies $\eta \leq 2\eta_{t+E}$ for $t \geq 1$.

Then, we have:

$$
\begin{aligned}
v &\leq 4\eta(2B + \varepsilon_{\max})^2(\gamma + 1) + \eta(\gamma + 1)C + (\gamma + 1)\Delta_1 \\
&= \eta(\gamma + 1)[4(2B + \varepsilon_{\max})^2 + C] + (\gamma + 1)\Delta_1
\end{aligned}
$$

Therefore,

$$
\begin{aligned}
\mathbb{E}[F_i(w_i^t)] - F_i^* &\leq \frac{L}{\gamma + t}[\eta(\gamma + 1)[4(2B + \varepsilon_{\max})^2 + C] + (\gamma + 1)\Delta_1] \\
&= \frac{\kappa}{\gamma + t}\left[\frac{\gamma + 1}{\gamma + t}(4(2B + \varepsilon_{\max})^2 + C) + \frac{\mu}{2}(\gamma + 1)\Delta_1\right] \\
&\leq \frac{\kappa}{\gamma + t}[4D + C + \frac{\mu}{2}(\gamma + 1)\Delta_1]
\end{aligned}
$$

where
$$
D = (2B + \varepsilon_{\max})^2
$$

$\square$

## D.4 Proof of the Key Lemmas

### D.4.1 Proof of Lemma 1

*Proof.* Given the local update rule:
$$
w_i^{t+1} = w_i^t - \eta\left(\nabla F_i(w_i^t, \xi_i^t) + g_i^t\right),
$$

our goal is to bound $\mathbb{E}\left[\|w_i^{t+1} - w_i^*\|^2\right]$

We start by expanding the squared norm:

$$\|w_i^{t+1} - w_i^*\|^2 = \left\|w_i^t - \eta\left(\nabla F_i(w_i^t, \xi_i^t) + g_i^t\right) - w_i^*\right\|^2$$
$$= \left\|w_i^t - w_i^* - \eta\left(\nabla F_i(w_i^t, \xi_i^t) + g_i^t\right)\right\|^2$$
$$= \|w_i^t - w_i^*\|^2 - 2\eta\left\langle \nabla F_i(w_i^t, \xi_i^t) + g_i^t, w_i^t - w_i^*\right\rangle + \eta^2\left\|\nabla F_i(w_i^t, \xi_i^t) + g_i^t\right\|^2$$

Taking expectations on both sides:

$$\mathbb{E}\left[\|w_i^{t+1} - w_i^*\|^2\right] = \mathbb{E}\left[\|w_i^t - w_i^*\|^2\right]$$
$$\underbrace{-2\eta\mathbb{E}\left[\left\langle \nabla F_i(w_i^t) + g_i^t, w_i^t - w_i^*\right\rangle\right]}_{A} + \underbrace{\eta^2\mathbb{E}\left[\left\|\nabla F_i(w_i^t, \xi_i^t) + g_i^t\right\|^2\right]}_{B}$$

We will bound $A$ and $B$ separately.

**Bounding $A$:**

We decompose $A$ into two parts:

$$A = \underbrace{-2\eta\mathbb{E}\left[\left\langle \nabla F_i(w_i^t), w_i^t - w_i^*\right\rangle\right]}_{A_1} \underbrace{-2\eta\mathbb{E}\left[\left\langle g_i^t, w_i^t - w_i^*\right\rangle\right]}_{A_2}$$

**Bounding $A_1$ using $\mu$-strong convexity:**

Since $F_i$ is $\mu$-strongly convex (Assumption 1), we have:

$$F_i(v) \geq F_i(w) + \nabla F_i(w)^\top(v - w) + \frac{\mu}{2}\|v - w\|^2, \quad \forall v, w$$

Let $v = w_i^*$ (the minimizer of $F_i$) and $w = w_i^t$. Then:

$$F_i(w_i^*) \geq F_i(w_i^t) + \nabla F_i(w_i^t)^\top(w_i^* - w_i^t) + \frac{\mu}{2}\|w_i^* - w_i^t\|^2$$
$$\Rightarrow \nabla F_i(w_i^t)^\top(w_i^t - w_i^*) \geq F_i(w_i^t) - F_i(w_i^*) + \frac{\mu}{2}\|w_i^t - w_i^*\|^2$$

Therefore,

$$A_1 = -2\eta\mathbb{E}\left[\nabla F_i(w_i^t)^\top(w_i^t - w_i^*)\right] \leq -2\eta\left(\mathbb{E}[F_i(w_i^t)] - F_i(w_i^*) + \frac{\mu}{2}\mathbb{E}\left[\|w_i^t - w_i^*\|^2\right]\right)$$

**Bounding $A_2$:**

Applying the Cauchy-Schwarz inequality:

$$A_2 = -2\eta\mathbb{E}\left[\left\langle g_i^t, w_i^t - w_i^*\right\rangle\right] \leq 2\eta\mathbb{E}\left[\|g_i^t\| \cdot \|w_i^t - w_i^*\|\right]$$

**Combining the bounds for $A_1$ and $A_2$:**

Thus,

$$A \leq -2\eta\left(\mathbb{E}[F_i(w_i^t)] - F_i(w_i^*)\right) - \eta\mu\mathbb{E}\left[\|w_i^t - w_i^*\|^2\right] + 2\eta\mathbb{E}\left[\|g_i^t\| \cdot \|w_i^t - w_i^*\|\right]$$

**Bounding $B$:**

We expand $B$:

$$B = \eta^2\mathbb{E}\left[\left\|\nabla F_i(w_i^t, \xi_i^t) + g_i^t\right\|^2\right]$$
$$= \underbrace{\eta^2\mathbb{E}\left[\left\|\nabla F_i(w_i^t, \xi_i^t)\right\|^2\right]}_{B_1} + \underbrace{\eta^2\mathbb{E}\left[\left\|g_i^t\right\|^2\right]}_{B_2} + \underbrace{2\eta^2\mathbb{E}\left[\left\langle \nabla F_i(w_i^t, \xi_i^t), g_i^t\right\rangle\right]}_{B_3}$$

**Bounding $B_1$:**

Using Assumption 3 and Assumption 4 (bounded variance and bounded gradient norm), we have:

$$B_1 = \eta^2 \mathbb{E}\left[\left\|\nabla F_i(w_i^t, \xi_i^t) - \nabla F_i(w_i^t) + \nabla F_i(w_i^t)\right\|^2\right]$$

$$\leq \eta^2\left(\mathbb{E}\left[\left\|\nabla F_i(w_i^t, \xi_i^t) - \nabla F_i(w_i^t)\right\|^2\right] + \left\|\nabla F_i(w_i^t)\right\|^2\right)$$

$$\leq \eta^2(\sigma^2 + G^2)$$

**Bounding $B_3$:**

Again, using the Cauchy-Schwarz inequality and Assumption 4:

$$B_3 = 2\eta^2 \mathbb{E}\left[\langle \nabla F_i(w_i^t, \xi_i^t), g_i^t\rangle\right] \leq 2\eta^2 \mathbb{E}\left[\left\|\nabla F_i(w_i^t, \xi_i^t)\right\| \cdot \left\|g_i^t\right\|\right]$$

$$\leq 2\eta^2 G \mathbb{E}\left[\left\|g_i^t\right\|\right]$$

since $\|\nabla F_i(w_i^t, \xi_i^t)\| \leq G$ (Assumption 4).

**Combining the bounds for $B_1$, $B_2$, and $B_3$:**

Therefore,

$$B \leq \eta^2(\sigma^2 + G^2) + \eta^2 \mathbb{E}\left[\left\|g_i^t\right\|^2\right] + 2\eta^2 G \mathbb{E}\left[\left\|g_i^t\right\|\right]$$

**Combining the bounds for $A$ and $B$:**

Substituting the bounds for $A$ and $B$ back into the main expression:

$$\mathbb{E}\left[\|w_i^{t+1} - w_i^*\|^2\right] \leq \mathbb{E}\left[\|w_i^t - w_i^*\|^2\right] - 2\eta\left(\mathbb{E}[F_i(w_i^t)] - F_i(w_i^*)\right) - \eta\mu\mathbb{E}\left[\|w_i^t - w_i^*\|^2\right]$$

$$+ 2\eta\mathbb{E}\left[\|g_i^t\| \cdot \|w_i^t - w_i^*\|\right] + \eta^2(\sigma^2 + G^2)$$

$$+ \eta^2\mathbb{E}\left[\left\|g_i^t\right\|^2\right] + 2\eta^2 G \mathbb{E}\left[\left\|g_i^t\right\|\right]$$

**Bounding $-2\eta\left(\mathbb{E}[F_i(w_i^t)] - F_i(w_i^*)\right)$:**

Using the $L$-smoothness of $F_i$ (Assumption 2), we have:

$$F_i(v) \leq F_i(w) + \nabla F_i(w)^\top(v - w) + \frac{L}{2}\|v - w\|^2, \quad \forall v, w$$

Let $v = w_i^*$ and $w = w_i^t$, then:

$$F_i(w_i^*) \leq F_i(w_i^t) + \nabla F_i(w_i^t)^\top(w_i^* - w_i^t) + \frac{L}{2}\|w_i^* - w_i^t\|^2$$

Rewriting:

$$F_i(w_i^t) - F_i(w_i^*) \geq \nabla F_i(w_i^t)^\top(w_i^t - w_i^*) - \frac{L}{2}\|w_i^t - w_i^*\|^2$$

Since $\nabla F_i(w_i^*) = 0$ (as $w_i^*$ minimizes $F_i$), and using the Cauchy-Schwarz inequality:

$$\left|\nabla F_i(w_i^t)^\top(w_i^t - w_i^*)\right| = \left|\left(\nabla F_i(w_i^t) - \nabla F_i(w_i^*)\right)^\top(w_i^t - w_i^*)\right|$$

$$\leq \left\|\nabla F_i(w_i^t) - \nabla F_i(w_i^*)\right\| \cdot \left\|w_i^t - w_i^*\right\|$$

$$\leq L\|w_i^t - w_i^*\|^2$$

Therefore,

$$F_i(w_i^t) - F_i(w_i^*) \geq -L\|w_i^t - w_i^*\|^2 - \frac{L}{2}\|w_i^t - w_i^*\|^2 = -\frac{3L}{2}\|w_i^t - w_i^*\|^2$$

Thus,

$$-2\eta\left(\mathbb{E}[F_i(w_i^t)] - F_i(w_i^*)\right) \leq 3\eta L \mathbb{E}\left[\|w_i^t - w_i^*\|^2\right]$$

**Final Combination**:

Substituting back into the main inequality:

$$\mathbb{E}\left[\|w_i^{t+1} - w_i^*\|^2\right] \leq \mathbb{E}\left[\|w_i^t - w_i^*\|^2\right] - \eta\mu\mathbb{E}\left[\|w_i^t - w_i^*\|^2\right] + 3\eta L\mathbb{E}\left[\|w_i^t - w_i^*\|^2\right]$$
$$+ 2\eta\mathbb{E}\left[\|g_i^t\| \cdot \|w_i^t - w_i^*\|\right] + \eta^2(\sigma^2 + G^2) + \eta^2\mathbb{E}\left[\|g_i^t\|^2\right]$$
$$+ 2\eta^2 G\mathbb{E}\left[\|g_i^t\|\right]$$

Simplifying the coefficients:

$$-\eta\mu + 3\eta L = \eta(-\mu + 3L)$$

Under the assumptions $\eta \leq \frac{1}{L}$ and $L \leq \frac{2}{3}$, we have

$$\mathbb{E}\left[\|w_i^{t+1} - w_i^*\|^2\right] \leq (1 - \eta\mu + 2\eta)\mathbb{E}\left[\|w_i^t - w_i^*\|^2\right] + \eta^2(\sigma^2 + G^2)$$
$$+ \eta^2\mathbb{E}\left[\|g_i^t\|^2\right] + 2\eta\mathbb{E}\left[\|g_i^t\| \cdot \|w_i^t - w_i^*\|\right] + 2\eta^2 G\mathbb{E}\left[\|g_i^t\|\right]$$

This completes the proof of Lemma 1. $\square$

### D.4.2 PROOF OF LEMMA 2

*Proof.* Given that the global momentum is defined as:

$$g_i^t = w_i^t - w_g^t,$$

where the global model $w_g^t$ is the weighted average of the locally compressed models:

$$w_g^t = \sum_{j=1}^{N} p_j C(w_j^t),$$

with $p_j$ being the proportion of data held by client $j$, and $C(w_j^t)$ representing the compressed model of client $j$.

Define the compression error for client $j$ as:

$$\varepsilon_j = C(w_j^t) - w_j^t.$$

Thus, the compressed model can be expressed as:

$$C(w_j^t) = w_j^t + \varepsilon_j.$$

Substituting back into the expression for $g_i^t$, we have:

$$g_i^t = w_i^t - \sum_{j=1}^{N} p_j \left(w_j^t + \varepsilon_j\right)$$
$$= w_i^t - \sum_{j=1}^{N} p_j w_j^t - \sum_{j=1}^{N} p_j \varepsilon_j.$$

Applying the triangle inequality to bound $\|g_i^t\|$:

$$\|g_i^t\| \leq \underbrace{\left\|w_i^t - \sum_{j=1}^{N} p_j w_j^t\right\|}_{D} + \underbrace{\left\|\sum_{j=1}^{N} p_j \varepsilon_j\right\|}_{E}.$$

We will bound each term separately.

**Bounding $D$**:

By definition,

$$D_t^i = \left\| w_i^t - \sum_{j=1}^N p_j w_j^t \right\|.$$

Using the triangle inequality and Assumption 5 (which states that $\|w_j^t\| \le B$ for all $j$):

$$D_t^i \le \left\| w_i^t \right\| + \left\| \sum_{j=1}^N p_j w_j^t \right\| \le B + \sum_{j=1}^N p_j \left\| w_j^t \right\| \le B + \sum_{j=1}^N p_j B$$

$$= B + B \left( \sum_{j=1}^N p_j \right) = B + B = 2B.$$

**Bounding $E$:**

Using the convexity of the norm and the fact that $\sum_{j=1}^N p_j = 1$:

$$\left\| \sum_{j=1}^N p_j \varepsilon_j \right\| \le \sum_{j=1}^N p_j \|\varepsilon_j\| \le \max_j \|\varepsilon_j\|$$

**Bounding $\max_j \|\varepsilon_j\|$:**

The compression error $\varepsilon_j$ arises from the weight clustering process applied to each layer of the neural network. The clustering aims to minimize the within-cluster sum of squares. Suppose a layer has $n_m$ weights (for layer $m$), and we partition them into $K_{\min}$ clusters.

In the worst-case scenario, all weights lie within a hypersphere of diameter $d$. The clustering algorithm divides this hypersphere into $K_{\min}$ approximately equal clusters. The diameter of each cluster is at most:

$$\delta_m = \frac{d}{K_{\min}^{1/\dim_m}},$$

where $\dim_m$ is the dimension of the weight space for layer $m$.

For any weight $w$ in layer $m$, the maximum distance between $w$ and its compressed value $C(w)$ is half the diameter of the cluster:

$$\|C(w) - w\| \le \frac{\delta_m}{2} = \frac{d}{2K_{\min}^{1/\dim_m}}.$$

Assuming $d \le 2B$ since Assumption 5, we have:

$$\|\varepsilon_j\| = \max_m \|C(w_j^{t,m}) - w_j^{t,m}\| \le \max_m \left\{ \frac{B}{K_{\min}^{1/\dim_m}} \right\}.$$

In onder to simplify our notation, let's define the maximum possible compression error:

$$\varepsilon_{\max} = \max_m \left\{ \frac{B}{K_{\min}^{1/\dim_m}} \right\}.$$

Therefore, the maximum compression error across all clients is bounded by:

$$\max_j \|\varepsilon_j\| \le \varepsilon_{\max}$$

**Final Bound on $\|g_i^t\|$:**

Combining the bounds for $D_t^i$ and $\varepsilon_{\max}$, we have:

$$\|g_i^t\| \leq D_t^i + \varepsilon_{\max}$$
$$\leq 2B + \varepsilon_{\max}.$$

**Conclusion**:

Since the bound holds for all $g_i^t$, taking expectations yields:

$$\mathbb{E}\left[\|g_i^t\|\right] \leq 2B + \varepsilon_{\max}.$$

This completes the proof of Lemma 2. $\qquad\square$

# E  SUPPLEMENTARY EXPERIMENTS

## E.1  ABLATION STUDY

### E.1.1  EFFECTIVENESS OF WEIGHT CLUSTERING PRUNING AND ADAPTIVE MECHANISM

We conducted an ablation study to evaluate the effectiveness of our proposed Weight Clustering Pruning (WCP) and the adaptive mechanism in AdFedWCP. We compared FedWCP (Federated Weighted Clustering Pruning with Fixed Number of Centroids) with different numbers of clusters $K$, AdFedWCP, and FedWCP (w/o WCP), which is our method without Weight Clustering Pruning. Both classification accuracy and communication overhead were analyzed, and the results are presented in Tables 3 and 4.

| Method | CIFAR-10 | EMNIST | CIFAR-100 |
|---|---|---|---|
| FedWCP ($K = 8$) | 60.56 | 83.89 | 19.84 |
| FedWCP ($K = 16$) | 61.96 | 85.46 | 20.36 |
| FedWCP ($K = 32$) | 63.38 | 85.93 | 20.96 |
| FedWCP (w/o WCP) | 63.66 | 85.99 | 22.72 |
| **AdFedWCP** | 61.04 | 85.12 | 20.44 |

Table 3: Top-1 clients test datasets average accuracy (%) of our methods on various datasets

| Method | LeafCNN (EMNIST) | LeNet (CIFAR-10) |
|---|---|---|
| FedWCP ($K = 8$) | 90.53% | 90.50% |
| FedWCP ($K = 16$) | 87.41% | 87.36% |
| FedWCP ($K = 32$) | 84.29% | 84.21% |
| FedWCP (w/o WCP) | 0% | 0% |
| **AdFedWCP** | 87.54% | 87.82% |

Table 4: Communication overhead reduction rates of our methods

From Table 3, it is evident that the classification accuracy of FedWCP ($K = 32$) remains similar to that of FedWCP (w/o WCP) across all datasets. This observation indicates that WCP can effectively reduce communication overhead without significantly impacting model performance. As the number of clusters $K$ increases, the accuracy of FedWCP improves; however, Table 4 shows that the communication overhead also increases. This trend demonstrates a trade-off between model performance and communication efficiency.

Moreover, AdFedWCP achieves competitive accuracy without the need for manual selection of $K$, while maintaining high compression rates. For instance, on the EMNIST dataset, AdFedWCP attains an accuracy of 85.12% with a compression rate of 87.54%, effectively balancing accuracy and communication efficiency.

Table 4 highlights the communication compression rates of our methods relative to FedWCP (w/o WCP), which serves as the baseline with no compression. Our methods achieve substantial communication savings compared to the baseline. For example, FedWCP ($K = 8$) reduces communication overhead by 90.53% on the LeafCNN model. However, a smaller $K$ leads to higher compression

rates but may slightly reduce accuracy, as observed with FedWCP ($K = 8$). In contrast, AdFedWCP balances communication efficiency and accuracy, achieving high compression rates and competitive accuracy without the need for manual tuning.

These results validate the effectiveness of our WCP strategy and the adaptive mechanism in AdFed-WCP for reducing communication overhead while maintaining high model performance, demonstrating a favorable trade-off between communication efficiency and model performance.

### E.1.2 IMPACT OF LAYER IMPORTANCE ESTIMATION

To further understand the effectiveness of layer importance estimation in our adaptive model, we conducted a supplemental ablation study specifically focused on the integration of the Imprinting method into AdFedWCP. This experiment compared the standard AdFedWCP configuration with and without the utilization of layer importance estimation.

| Method | Accuracy | Communication overhead reduction rates |
|---|---|---|
| AdFedWCP (w/o imprinting) | 85.02% | 97.29% |
| **AdFedWCP** | 85.12% | 87.54% |

Table 5: Comparison of AdFedWCP with and without Layer Importance Estimation on EMNIST Dataset

From Table 5, it is evident that integrating layer importance estimation through the Imprinting method slightly enhances both accuracy and communication efficiency. AdFedWCP with Imprinting achieved a higher accuracy and compression rate compared to the version without it.

The increment in accuracy and compression rate with Imprinting integration suggests that this method is efficient at identifying and emphasizing layers that contribute most significantly to model performance. This approach not only ensures a more effective allocation of model resources but also assists in achieving a refined balance between model accuracy and communication overhead.

Moreover, the observed improvements underscore the value of precision in layer importance assessment within dynamic environments where computational resources and bandwidth are limited. By efficiently pinpointing crucial layers, the Imprinting method enhances the overall utility and effectiveness of the adaptive pruning mechanism in AdFedWCP.

These findings validate our hypothesis that layer importance estimation can significantly contribute to optimizing federated learning strategies by facilitating more informed and strategic model adjustments. This, in turn, affirms the necessity of incorporating sophisticated layer evaluation techniques in complex models, especially in scenarios characterized by data and environmental heterogeneity.

### E.2 EXPERIMENTS ON VARIOUS DATA HETEROGENEITY LEVELS

To evaluate the effectiveness of our proposed AdFedWCP method in handling different data heterogeneity environments, we conducted detailed comparative experiments. In this study, we considered various Dirichlet distribution parameters $\alpha$, adjusting the $\alpha$ value to simulate data heterogeneity conditions ranging from mild to extreme. We compare the AdFedWCP and FedWCP methods with other baseline methods, specifically including FedWCP with different numbers of centroids ( K ) and (w/o WCP). We selected three datasets: CIFAR-10, EMNIST, and CIFAR-100, and conducted experiments on each dataset using different $\alpha$ values (0.1, 0.4, 1). These $\alpha$ values represent different degrees of data heterogeneity, where $\alpha = 0.1$ indicates high heterogeneity, while $\alpha = 1$ indicates low data heterogeneity.

The results in Figure 6 show the performance of various methods in dealing with data with different degrees of heterogeneity. By comparing the experimental results under different Dirichlet distribution parameters $\alpha$ (0.1, 0.4, 1), we can draw the following main conclusions and analyses:

The impact of data heterogeneity on federated learning algorithms is evident as performance generally improves across all methods and datasets when $\alpha$ increases from 0.1 to 1, suggesting that higher data heterogeneity (lower $\alpha$) presents significant challenges. The performance gap between different methods becomes more pronounced under high heterogeneity ($\alpha = 0.1$), indicating that some methods are more robust to non-IID data distributions. Regarding the comparative performance

| Method | CIFAR-10 | | | EMNIST | | | CIFAR-100 | | |
|---|---|---|---|---|---|---|---|---|---|
| | $\alpha = 0.1$ | $\alpha = 0.4$ | $\alpha = 1$ | $\alpha = 0.1$ | $\alpha = 0.4$ | $\alpha = 1$ | $\alpha = 0.1$ | $\alpha = 0.4$ | $\alpha = 1$ |
| FedAvg | 52.91% | 60.64% | 63.57% | 68.68% | 81.83% | 83.79% | 12.36% | 18.75% | 26.48% |
| FedEM | 60.44% | 62.19% | 65.01% | 77.55% | 84.35% | 85.30% | 20.92% | 22.39% | 28.58% |
| q-FedCG | 47.79% | 52.45% | 62.32% | 51.71% | 80.24% | 75.94% | 9.64% | 13.50% | 15.92% |
| FedMask | 40.07% | 48.05% | 70.13% | 55.51% | 63.33% | 78.92% | 7.54% | 11.31% | 21.97% |
| pFedGate | 53.39% | 60.36% | 70.42% | 81.12% | 82.11% | 84.08% | 12.87% | 13.50% | 21.01% |
| FedWCP (K=8) | 52.97% | 60.56% | 67.71% | 79.37% | 83.89% | 89.79% | 12.01% | 19.84% | 36.29% |
| FedWCP (K=16) | 55.16% | 61.96% | 69.88% | 82.37% | 85.46% | 91.61% | 12.53% | 20.36% | 37.73% |
| FedWCP (K=32) | 58.09% | 63.58% | 71.57% | 83.01% | 85.93% | 92.08% | 13.91% | 20.96% | 38.89% |
| FedWCP (w/o WCP) | 59.16% | 63.66% | 71.54% | 84.73% | 85.99% | 93.29% | 16.45% | 22.72% | 39.25% |
| **AdFedWCP** | 54.48% | 61.04% | 68.86% | 82.00% | 85.12% | 90.28% | 13.10% | 20.44% | 39.19% |

Table 6: Top-1 test datasets accuracy of different methods on various datasets under different $\alpha$ values of the Dirichlet distribution

of methods, FedWCP (w/o WCP) consistently excels across all datasets and heterogeneity levels, often achieving the highest accuracy, particularly in scenarios of low heterogeneity ($\alpha = 1$). The FedWCP also shows strong performance, with accuracy generally improving as $K$ increases; the $K = 32$ variant frequently outperforms other methods, especially in moderate to low heterogeneity settings. AdFedWCP demonstrates competitive performance, often comparable to or surpassing FedWCP ($K = 32$), notably in CIFAR-100 with $\alpha = 1$. Meanwhile, FedEM maintains robust performance across different levels of heterogeneity, consistently outperforming FedAvg, whereas q-FedAvg and FedMask tend to underperform, particularly in scenarios of high heterogeneity. The effectiveness of AdFedWCP is highlighted by its comparable or sometimes superior performance to FedWCP, particularly in low heterogeneity scenarios and on more complex datasets like CIFAR-100. Its adaptive nature allows it to excel across various levels of heterogeneity without the need for manual tuning of $K$, offering a good balance between performance and practical applicability. However, while FedWCP (w/o WCP) often achieves the highest accuracy, it incurs higher computational or communication costs compared to methods like AdFedWCP or FedWCP. The choice between FedWCP variants and AdFedWCP may thus depend on the specific use case, with AdFedWCP offering adaptability and FedWCP with higher $K$ values potentially providing slightly better performance in some scenarios.

In conclusion, the experimental results highlight the effectiveness of clustering-based methods (Fed-WCP, AdFedWCP) and ensemble methods (FedWCP (w/o WCP), FedEM) in handling data heterogeneity in federated learning. AdFedWCP, in particular, demonstrates a good balance between performance and adaptability across different heterogeneity levels and datasets.

While communication efficiency is a crucial aspect of federated learning, our observations indicate that the communication overhead reduction achieved by AdFedWCP remains relatively consistent. As shown in Table 2, the communication overhead reduction rates for our method are significant and stable across various datasets and network architectures. Therefore, including additional experiments on communication efficiency under varying data heterogeneity levels would not provide further insights. Our focus in this section is to analyze how different methods handle data heterogeneity in terms of model performance, where variations are more pronounced and informative.

### E.3 SPARSITY OF MODEL PARAMETERS

This section aims to evaluate the sparsity of weight matrices after training for different federated learning methods to determine their effectiveness in reducing computational overhead. Pruning removes connections deemed unimportant by setting their corresponding weights to zero. This process results in sparse weight matrices, and leveraging the properties of sparse matrices can enhance computational efficiency and reduce memory usage (Xu & McAuley, 2023). By comparing the average sparsity rates achieved by each method, we can visually assess their performance in reducing model parameter complexity.

We conducted tests on two different network architectures (LeafCNN and LeNet). The following methods were compared: AdFedWCP (our proposed method that does not require manual setting of sparsity), FedWCP with $K = 8$, $K = 16$, $K = 32$ (different configurations of the FedWCP method with $K$ values of 8, 16, and 32), pFedGate, FedMask, qFedCG (all these methods manually set a 50% sparsity rate, with pFedGate capable of adaptive sparsity adjustment).

| Method | LeafCNN (EMNIST) | LeNet (CIFAR-10) |
|---|---|---|
| FedMask | 50.00% | 50.00% |
| pFedGate | 23.42% | 23.18% |
| FedWCP with $K = 8$ | 22.79% | 58.99% |
| FedWCP with $K = 16$ | 12.27% | 44.70% |
| FedWCP with $K = 32$ | 5.82% | 29.38% |
| **AdFedWCP** | 39.56% | 54.19% |

Table 7: Sparsity rates of different federated learning methods on two datasets

First, it was observed that AdFedWCP displayed considerable sparsity across different datasets and model architectures, indicating the effectiveness of its model pruning. Especially on the LeNet model, where AdFedWCP achieved a sparsity rate of 54.19%, it not only demonstrates robustness in handling complex datasets but also shows that it can effectively reduce the computational burden of the model without sacrificing performance. Moreover, the effectiveness of AdFedWCP's adaptive sparsity adjustment capability can be further attributed to its unique approach of dynamically determining the number of centroids per layer based on the importance of each layer. This tailored granularity ensures that less important layers utilize fewer centroids, which contributes to increased overall sparsity without compromising the performance of critical parts of the model.

Second, comparing different $K$ values of the FedWCP configuration, we found that as $K$ increased, the sparsity rate significantly decreased. For instance, in the LeafCNN model, the sparsity rate was 22.79% for $K = 8$ and only 5.82% for $K = 32$. This phenomenon reveals that larger $K$ values lead to denser weight matrices, which may increase the model's computational complexity—an important consideration for applications deployed on resource-constrained devices, as higher computational complexity could result in greater energy consumption and slower response times.

Furthermore, pFedGate and FedMask both had manually set sparsity rates of 50%. Although Fed-Mask simplifies the setting process for sparsity, it cannot dynamically adjust based on actual training situations. Such static setting methods may not be flexible enough.

Finally, the adaptive adjustment strategies of AdFedWCP and FedWCP offered higher flexibility and potential performance advantages. This adaptive capability is particularly suitable for the variable federated learning environment, where client data distributions and computational capabilities can vary widely. pFedGate also has an adaptive sparsity rate strategy but did not further optimize for communication overhead.

In conclusion, the AdFedWCP method, with its highly adaptive sparsity adjustment capability, not only confirms the effectiveness of its pruning strategy but also optimizes according to specific data characteristics and bandwidth resources, demonstrating broad applicability and effectiveness in real-world scenarios.

### E.4 Hyperparameter Studies

#### E.4.1 Impact of the Hyperparameter $\lambda$

The hyperparameter $\lambda$ plays a pivotal role in balancing the weight updates between the global and local models. Its importance lies in two aspects: enabling global knowledge sharing by integrating momentum information from the global model and enhancing personalized learning by maintaining consistency with the global model while preserving local data characteristics. To address the variations in data and model states across different training phases, we employed a loss-based annealing mechanism in our experiments to dynamically adjust $\lambda$. This mechanism improves training stability and efficiency, especially in heterogeneous data environments.

The annealing mechanism adjusts $\lambda$ dynamically based on the relationship between the current loss and the exponentially smoothed loss. Specifically, the exponential smoothing loss is computed as:

$$L_{\exp}^{(t)} = \alpha L^{(t)} + (1 - \alpha) L_{\exp}^{(t-1)}$$

where $L^{(t)}$ represents the current loss, and $\alpha$ is set to 0.5 in our experiments. The momentum decay factor $d^{(i)}$ is then adjusted as:

$$d^{(i)} = \begin{cases} \min(\text{base\_decay\_rate}^{i+1} \times 1.1, 0.8), & \text{if } L^{(t)} < L_{\exp}^{(t)} \\ \max(\text{base\_decay\_rate}^{i+1} \div 1.1, 0.1), & \text{otherwise} \end{cases}$$

This dynamic adjustment improves both global knowledge sharing in early training and local adaptation in later stages. Using the decay factor, the model update rule is defined as:

$$\nabla \theta_i^{(t)} = \nabla \theta_i^{(t)} + d^{(i)} \cdot \Delta \theta_{\text{ref}}$$

where $\Delta \theta_{\text{ref}}$ represents the difference between the global and local model parameters.

| Configuration | Average Accuracy |
|---|---|
| $\lambda = 0.1$ | 82.36 |
| $\lambda = 0.45$ | 70.26 |
| $\lambda = 0.8$ | 64.70 |
| Dynamic Adjustment (Annealing) | **85.12** |

Table 8: Impact of $\lambda$ on EMNIST accuracy(%)

To validate the effectiveness of $\lambda$ and the annealing mechanism, we conducted experiments with different fixed $\lambda$ values and dynamic adjustments. The results, shown in Table 8, highlight that a fixed $\lambda$ can either overly rely on global information (e.g., $\lambda = 0.8$) or overfit to local data (e.g., $\lambda = 0.1$), both leading to suboptimal performance. In contrast, the annealing mechanism achieves the highest accuracy of 85.12% on EMNIST by balancing global and local adaptation across different training stages.

### E.4.2 IMPACT OF THE HYPERPARAMETERS $\xi$ AND $\zeta$

The hyperparameters $\xi$ and $\zeta$ control the adjustment magnitude of the optimization lower bound when dynamically modifying the number of cluster centers. These parameters significantly affect the trade-off between model compression and accuracy. Specifically, $\xi$ prevents premature reduction of the optimization lower bound when the model improves, maintaining stability, while $\zeta$ accelerates recovery of the optimization lower bound during performance degradation.

The optimization lower bound $\eta$ is defined as:

$$\eta = \begin{cases} 1 - \xi \cdot |\Delta A|, & \text{if } \Delta A > 0 \\ 1 + \zeta \cdot |\Delta A|, & \text{if } \Delta A < 0 \end{cases}$$

where $\Delta A = A^{(t)} - A^{(t-1)}$ represents the change in model accuracy between the current and previous rounds. $A^{(t)}$ is the model accuracy at the $t$-th round.

Table 9 presents the experimental results on EMNIST for various combinations of $\xi$ and $\zeta$. Smaller $\xi$ values (e.g., $\xi = 0.1$

| $\xi$ | $\zeta$ | Accuracy | Communication overhead reduction rates |
|---|---|---|---|
| **0.1** | **1.5** | **85.12%** | **87.54%** |
| 0.5 | 1.5 | 84.96% | 87.47% |
| 1.0 | 1.5 | 85.00% | 87.30% |
| 0.1 | 1.0 | 84.96% | 87.55% |
| 0.5 | 1.0 | 84.95% | 87.60% |
| 1.0 | 1.0 | 84.92% | 87.59% |
| 0.1 | 0.5 | 84.98% | 87.61% |
| 0.5 | 0.5 | 85.02% | 87.65% |
| 1.0 | 0.5 | 84.79% | 87.66% |

Table 9: Impact of $\xi$ and $\zeta$ on EMNIST accuracy and compression rate

E.5    LEARNING CURVES

In response to the reviewers' suggestions, we have added supplementary learning curves for AdFed-WCP and baseline methods on two datasets: EMNIST and CIFAR-10. These curves illustrate the accuracy progression over multiple training rounds and emphasize the stability and performance advantages of AdFedWCP in heterogeneous environments. The learning curves are presented in Figure 2.

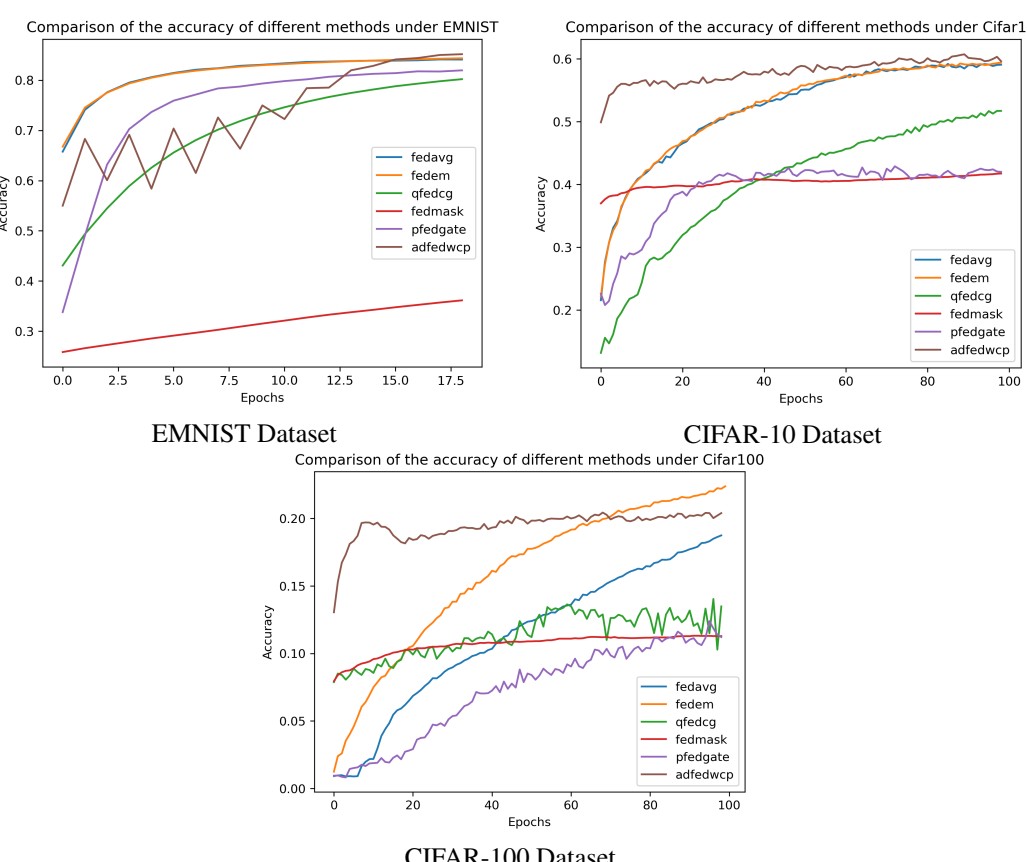

EMNIST Dataset

CIFAR-10 Dataset

CIFAR-100 Dataset

Figure 2: Learning curves comparing the accuracy of different methods on the EMNIST (a), CIFAR-10 (b), and CIFAR-100 (c) datasets.

E.5.1    ANALYSIS OF THE EMNIST LEARNING CURVES

On the EMNIST dataset, AdFedWCP demonstrates superior adaptability and stability compared to baseline methods. In the early training stages, AdFedWCP effectively balances global knowledge sharing with local personalized model adaptation. While initial fluctuations are observed due to significant data heterogeneity, the global model increasingly integrates client-specific characteristics as training progresses, leading to reduced fluctuations and improved stability.

In the later stages of training, AdFedWCP stabilizes at an accuracy of approximately 84%, outperforming baseline methods such as FedAvg and FedEM. Although FedAvg and FedEM achieve similar performance levels, AdFedWCP shows a clear advantage in handling heterogeneous environments. Additionally, pFedGate exhibits faster convergence in the early stages but falls significantly behind AdFedWCP in the final accuracy, highlighting its limitations under high heterogeneity. FedMask performs the worst, with limited learning capacity due to its communication-constrained design.

### E.5.2 ANALYSIS OF THE CIFAR-10 LEARNING CURVES

On the CIFAR-10 dataset, AdFedWCP similarly outperforms other baseline methods. In the early training rounds, AdFedWCP quickly converges to an accuracy of approximately 55%, showcasing its efficiency in adapting to image classification tasks. In the later stages, AdFedWCP achieves a final accuracy of approximately 61%, surpassing FedAvg, FedEM, and other methods.

FedMask again exhibits the poorest performance, stabilizing at an accuracy of only around 40%, reflecting its limited adaptability under communication-constrained conditions. While FedAvg and FedEM converge to acceptable accuracy levels, AdFedWCP consistently demonstrates better convergence behavior and adaptability to the heterogeneous and challenging CIFAR-10 dataset.

These results validate the robustness and efficiency of AdFedWCP in handling diverse datasets and highlight its ability to balance global and local knowledge, ensuring strong performance across varying degrees of heterogeneity.

### E.5.3 ANALYSIS OF THE CIFAR-100 LEARNING CURVES

On the CIFAR-100 dataset, AdFedWCP shows significant prowess, surpassing other baseline federated learning methods in both adaptability and stability. In the initial stages of training, AdFedWCP experiences minor fluctuations, likely due to the inherent data heterogeneity within the dataset. However, it swiftly demonstrates its capability to integrate diverse client-specific characteristics, enhancing the global model's accuracy while maintaining consistent performance.

As training progresses, AdFedWCP continues to excel, ultimately stabilizing at a notably high accuracy level compared to other methods. This superior performance illustrates AdFedWCP's effective balancing of global knowledge sharing with local model optimization, even in a complex and diverse data environment like CIFAR-100. The method's final accuracy not only exceeds that of FedAvg and FedEM, which demonstrate moderate performance improvements, but also significantly outperforms qFedCG, FedMask, and pFedGate. These latter methods show either excessive fluctuations or slower convergence rates, indicating possible challenges in handling high heterogeneity or limitations in learning capacity under CIFAR-100's extensive class variety.

### E.6 COMPARISON WITH FEDKD

We have conducted supplementary experiments to strengthen the evaluation of AdFedWCP. While we appreciate the importance of comparative analysis, we would like to clarify that AdFedWCP and FedKD Wu et al. (2022) target different research objectives and operate under distinct application scenarios. Below, we detail these differences and present the results of supplementary experiments.

### E.6.1 DIFFERENCES IN RESEARCH OBJECTIVES BETWEEN ADFEDWCP AND FEDKD

FedKD primarily focuses on reducing communication costs through knowledge distillation, addressing scenarios where heterogeneous model architectures are employed across clients. This method aims to handle the challenges of collaboration and communication when clients have diverse neural network architectures.

AdFedWCP, by contrast, is designed to tackle data and bandwidth heterogeneity by dynamically clustering and pruning model weights. This approach assumes consistent model architectures across clients and prioritizes communication efficiency while maintaining high model performance under diverse data and bandwidth distributions.

Given these differing assumptions, direct comparisons between FedKD and AdFedWCP may not fully capture the respective strengths of the two methods. However, to provide a quantitative evaluation, we designed experiments that align with AdFedWCP's assumptions.

### E.6.2 EXPERIMENTAL SETUP

To ensure fairness, we adjusted the experimental setup. Since AdFedWCP assumes identical architectures across clients, we configured all clients in FedKD to also use a homogeneous model architecture. In FedKD, global knowledge distillation was based on models trained with identical architectures to maintain consistency with the AdFedWCP framework. These adjustments allowed

us to fairly evaluate both methods under the same conditions, highlighting their respective performance in scenarios with homogeneous models and heterogeneous data distributions.

### E.6.3 Experimental Results and Analysis

| Method | Accuracy | Communication overhead reduction rates |
|--------|----------|----------------------------------------|
| FedKD | 84.06% | 73.18% |
| **AdFedWCP** | **85.12%** | **87.54%** |

Table 10: Comparison of AdFedWCP and FedKD on EMNIST.

The experimental results on the EMNIST dataset are summarized in Table 10. AdFedWCP demonstrated clear advantages in both accuracy and communication compression rate compared to FedKD. AdFedWCP achieved an accuracy of 85.12%, surpassing FedKD's 84.06%. This demonstrates that dynamic weight clustering and pruning is more effective in handling data heterogeneity, thereby improving model performance in federated learning. AdFedWCP significantly outperformed FedKD in communication compression, achieving a compression rate of **87.54%** compared to FedKD's **73.18%**. This highlights the efficiency of AdFedWCP's dynamic pruning mechanism in reducing communication overhead, particularly in bandwidth-constrained environments.

The results illustrate that while FedKD effectively compresses communication through knowledge distillation, it is not explicitly optimized for bandwidth heterogeneity, which may limit its applicability in such scenarios. In contrast, AdFedWCP's dynamic adjustment mechanism is specifically designed to address bandwidth and data heterogeneity, making it more suitable for environments where these challenges are prevalent.

### E.7 Performance Evaluation Under Extreme Heterogeneity

To further assess the adaptability of AdFedWCP in highly heterogeneous scenarios, we conducted experiments to evaluate its performance under varying client bandwidth conditions. Clients were divided into five distinct groups based on their bandwidth ranges, simulating environments with diverse communication capabilities. The bandwidth groups were categorized as follows: 5 Mbps - 24 Mbps, 24 Mbps - 43 Mbps, 43 Mbps - 62 Mbps, 62 Mbps - 81 Mbps, and 81 Mbps - 100 Mbps. The results are shown in Figure 3.

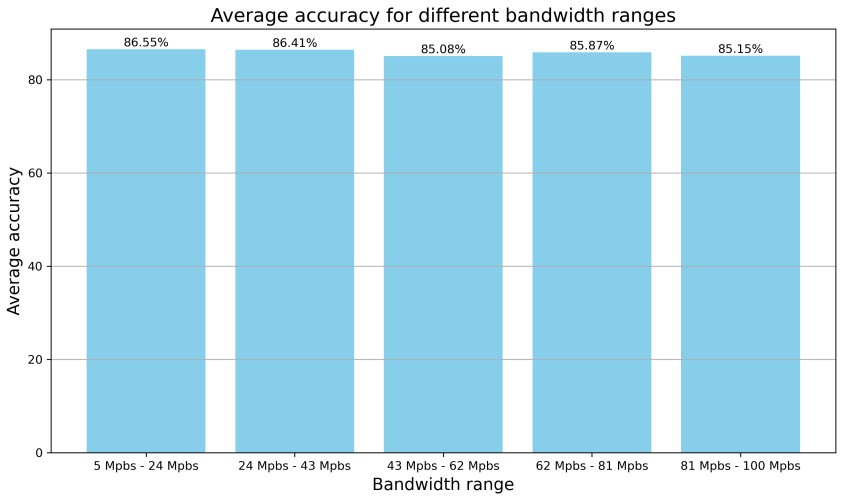

Figure 3: Average accuracy of AdFedWCP across different bandwidth ranges.

The experimental results demonstrate that the accuracy of AdFedWCP varies minimally across different bandwidth groups, with a difference of less than 1.5%. Specifically, even in the lowest bandwidth group (5 Mbps - 24 Mbps), AdFedWCP achieves an average accuracy of 86.55%. This stability highlights the robustness of the dynamic weight clustering and pruning mechanism, which

effectively balances communication efficiency and model performance. By dynamically adjusting pruning rates and model complexity to adapt to each client's environment, AdFedWCP ensures high accuracy even in extreme bandwidth conditions. These findings validate the effectiveness of AdFedWCP in addressing bandwidth heterogeneity, maintaining consistent and efficient model performance across diverse communication environments.

## E.8 THE ROLE OF UPPER AND LOWER BOUNDS

To further explore the trade-off between communication efficiency and model performance, we conducted ablation studies analyzing the effects of the upper and lower bounds in AdFedWCP. These bounds regulate the pruning rates to balance accuracy and compression. We evaluated three configurations: removing the lower bound, removing the upper bound, and retaining both bounds. The results are presented in Table 11.

| Configuration | Accuracy (%) | Compression Rate (%) |
|---|---|---|
| Without Lower Bound | 78.27% | 90.53% |
| Without Upper Bound | 85.12% | 87.37% |
| **With Both Bounds (Default)** | **85.12%** | **87.54%** |

Table 11: Impact of removing upper and lower bounds on accuracy and compression rate.

The experimental results demonstrate the critical role of both the upper and lower bounds in AdFedWCP:

When the lower bound is removed, the accuracy drops significantly to 78.27%, while the compression rate increases to 90.53%. This indicates that excessive pruning without the lower bound leads to higher communication savings but severely degrades model performance. The lower bound thus plays a vital role in preserving accuracy by preventing over-pruning.

Conversely, removing the upper bound has no impact on accuracy (remaining at 85.12%), but the compression rate decreases to 87.37%. This reflects insufficient pruning, which increases communication overhead. The upper bound is therefore essential for maintaining communication efficiency by controlling excessive communication costs.

With both bounds enabled, AdFedWCP achieves the optimal balance between accuracy and compression, with an accuracy of 85.12% and a compression rate of 87.54%. These results validate that the upper and lower bounds are crucial components in the design of AdFedWCP, effectively balancing communication efficiency and model performance in heterogeneous federated learning scenarios.

## F POTENTIAL LIMITATIONS OF ADFEDWCP

While AdFedWCP demonstrates strong performance in addressing bandwidth heterogeneity and improving communication efficiency, there are several limitations that warrant consideration for future research and practical applications. First, AdFedWCP assumes static communication bandwidths for clients throughout the training process. While this simplifies the experimental setup and enables controlled evaluation, real-world federated learning systems often encounter dynamic bandwidth fluctuations. Incorporating mechanisms to address dynamic bandwidth changes could further enhance the robustness and adaptability of AdFedWCP in practical deployments.

Second, AdFedWCP uses the Imprinting method for layer importance evaluation, which is computationally efficient and well-suited for resource-constrained federated learning environments. However, this approach limits the evaluation to a single method, leaving the potential benefits of other importance evaluation techniques unexplored. Future work could investigate alternative methods, such as saliency-based or gradient-based techniques, to optimize the pruning strategy further and improve model performance.

Lastly, AdFedWCP primarily focuses on addressing bandwidth heterogeneity among clients, with limited consideration for computational heterogeneity, such as differences in processing power or memory capacity across devices. While the weight clustering pruning mechanism has the potential

to reduce computational overhead by creating sparse matrices, the sparsity generated by clustering may not be structured. As a result, the improvement with computational efficiency is not as significant as that of structured sparsity methods. Extending AdFedWCP to explicitly address computational heterogeneity or structured sparsity could significantly enhance its applicability in highly resource-constrained environments.

These limitations suggest clear directions for future work, including the development of strategies to handle dynamic bandwidth, exploration of alternative layer importance evaluation methods, and explicit optimization for computational heterogeneity. Addressing these challenges could further enhance the adaptability, scalability, and efficiency of AdFedWCP in diverse and practical federated learning scenarios.

## G  VISUAL REPRESENTATION OF THE ADFEDWCP WORKFLOW

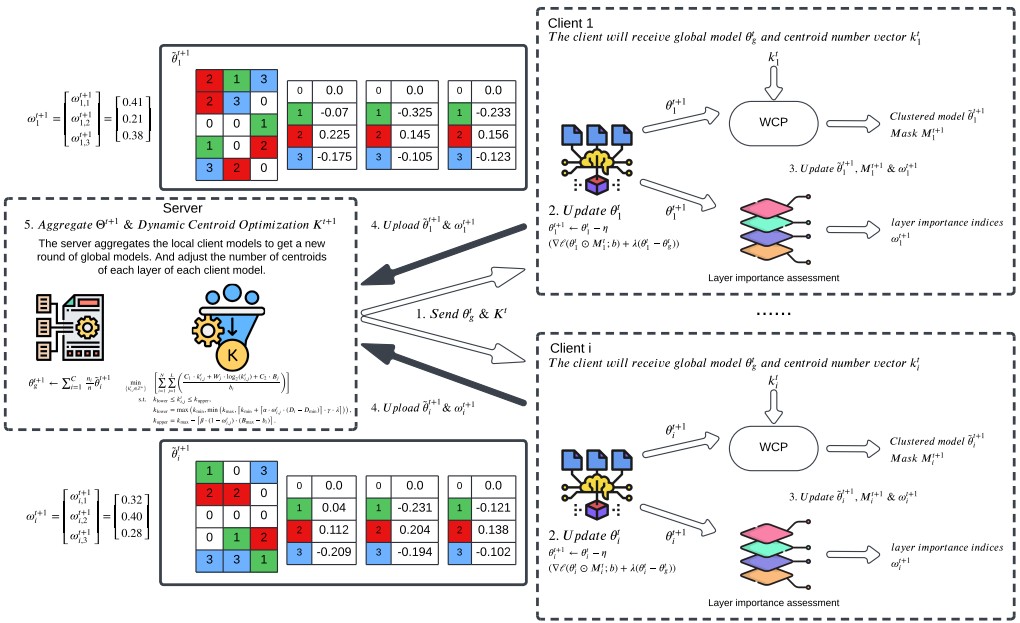

Figure 4: Workflow of AdFedWCP: Dynamic Weight Clustering Pruning with Adaptive Centroid Optimization.

This figure illustrates the complete workflow of the AdFedWCP (Adaptive Federated Weight Clustering Pruning) framework, showcasing the dynamic interaction between the server and the clients. Initially, the server broadcasts the global model parameters $\theta_g^t$ and the centroid number vector $k^t$ to all clients. Each client then updates its local model by applying Weight Clustering Pruning (WCP) to produce a pruned model $\tilde{\theta}_i^{t+1}$ and updates the model based on the local data characteristics and the received global model parameters. This step includes generating a pruning mask and assessing the layer importance, which guides the dynamic adjustment of the centroid numbers.

Subsequently, clients upload their pruned models and the corresponding layer importance indices back to the server. The server aggregates these models to update the global model $\theta_g^{t+1}$ and dynamically optimizes the number of centroids for the next iteration based on the aggregated layer importance indices and client-specific constraints. This dynamic centroid optimization is aimed at balancing the computational load and communication overhead across heterogeneous network conditions, thus enhancing the overall efficiency and effectiveness of the federated learning process.