# OpenReview forum: "Efficient Personalized Federated Learning via Adaptive Weight Clustering Pruning"
_ICLR.cc/2025/Conference — Submitted to ICLR 2025_

### Official Review · Reviewer_sERM · 2024-10-29

**Soundness:** 2
**Presentation:** 2
**Contribution:** 2
**Rating:** 5
**Confidence:** 4

**Summary:**

This paper proposes AdFedWCP framework, aims to improve personalized federated learning in the face of local data and communication bandwidth heterogeneity while maintaining model performance and communication efficiency. AdFedWCP features a adaptive weight clustering pruning strategy to reduce the per-client model size based on each client’s characteristics. This mechanism dynamically adjusts cluster centroids per layer based on layer importance, client data, and communication constraints through an integrated adaptive scheme, effectively addressing bandwidth heterogeneity.

**Strengths:**

1.	Convergence analysis of the proposed method is provided.
2.	The proposed method achieves significant communication overhead reduction.

**Weaknesses:**

1.	The aim of this paper is to propose an efficient FL approach “for resource-constrained federated learning environments”. However, the main contribution of this paper is limited to the communication cost, which is insufficient to address the resource constraints of heterogeneous clients. It is not clear whether the overall resource consumption such as computational overhead will be reduced.
2.	In this paper, weight clustering and layer importance estimation are integrated, but the unique contributions of these two strategies compared to existing work are not clear. The novelty of simply converting these centralized strategies to distributed versions is limited.
3.	In this paper, three key components are designed for AdFedWCP: weight clustering pruning, layer importance estimation, and dynamic centroid optimization strategies. However, this paper does not discuss their resource consumption, which is important for efficient FL methods.
4.	The ablation study in this paper could be improved, especially the empirical results about resource consumption of weight clustering pruning, layer importance estimation and dynamic centroid optimization strategies.

**Questions:**

1. Is AdFedWCP applicable to heterogeneous clients with different resource capabilities (including computing resources, communication resources, storage resources, etc.)?
2. How does AdFedWCP handle the situation where the clients in the FL cannot afford the resource requirements of the weight clustering pruning, layer importance estimation, and dynamic centroid optimization policies?

---

> ### Author Response · Authors · 2024-11-23
> **Response to Weakness 1**
>
> ### On whether AdFedWCP is applicable to heterogeneous clients with varying resource capacities (Weakness 1)
>
> We appreciate the reviewer’s insightful question. AdFedWCP is primarily designed to address the challenges of **data and communication bandwidth heterogeneity** in personalized federated learning. It is not specifically tailored to handle all types of resource constraints, such as computational and storage heterogeneity. The core objective of AdFedWCP is to optimize communication efficiency and adapt to bandwidth heterogeneity, ensuring robust model performance in environments with limited bandwidth and significant data distribution differences.
>
> That said, through the dynamic weight clustering and pruning mechanism, AdFedWCP inherently reduces model computational requirements to some extent, alleviating computational burdens for certain clients. This makes it applicable to most common federated learning scenarios. However, AdFedWCP does not explicitly target the challenges of computational or storage resource heterogeneity, as these are beyond the method’s primary research focus.

---

> ### Author Response · Authors · 2024-11-23
> **Response to Weakness 2**
>
> ### Explanation of the uniqueness and contribution of Weight Clustering and Layer Importance Evaluation Strategies (Weakness 2)
>
> We thank the reviewer for raising this question. Below, we provide a detailed explanation of the unique contributions of the weight clustering and layer importance evaluation strategies in AdFedWCP, along with comparisons to existing methods such as DKM [1].
>
> #### **1. Uniqueness of the Weight Clustering Pruning Strategy**
>
> The weight clustering pruning strategy in AdFedWCP is specifically designed for federated learning environments. Its uniqueness lies in the following aspects:
> - **Adaptation to Dynamic Heterogeneity in Federated Learning**:
>   Unlike traditional centralized model compression methods (e.g., DKM [1]), AdFedWCP's weight clustering pruning mechanism adaptively adjusts the pruning strategy:
>   - **Dynamic Adjustment of Cluster Centers**:
>     The clustering strategy dynamically optimizes the number of cluster centers based on each client’s specific bandwidth and data distribution, addressing the prevalent heterogeneity in federated learning.
>   - **Communication Efficiency-First Design**:
>     AdFedWCP directly reduces communication costs by pruning model parameters, prioritizing bandwidth constraints rather than solely optimizing model performance.
>
> - **Core Innovations Compared to Traditional Centralized Methods**:
>   - Centralized methods like DKM typically operate on a single server, accessing the full model data and optimizing cluster centers using gradient-based approaches.
>   - AdFedWCP, in contrast, is designed for the distributed nature of federated learning, with a client-driven dynamic pruning strategy better suited for practical scenarios.
>   - AdFedWCP's dynamic adaptive design achieves an effective balance between performance and communication efficiency under varying client bandwidth and data characteristics.
>
> #### **2. Uniqueness of Layer Importance Evaluation**
>
> The layer importance evaluation in AdFedWCP is distinctive in the following ways:
> - **Efficiency of the Method**:
>   AdFedWCP uses the Imprinting method for layer importance evaluation, which requires only a single forward pass to compute layer importance. Compared to other methods that involve multiple iterations or complex optimizations, this approach significantly reduces computational overhead, making it ideal for resource-constrained federated learning clients.
>
> - **Guidance for Dynamic Pruning Strategy**:
>   - The results of layer importance evaluation directly influence the design of the pruning strategy, allowing the pruning intensity to be dynamically allocated based on the importance of each layer.
>   - This strategy not only enhances pruning efficiency in federated learning but also achieves more fine-grained model compression optimization across different layers.
>
> - **Applicability in Federated Learning Environments**:
>   The simplicity and efficiency of the Imprinting method make it an ideal choice for resource-constrained clients in federated learning, while still providing effective guidance for pruning strategies.
>
> #### **3. Key Differences from Existing Methods (e.g., DKM [1])**
> Although AdFedWCP’s strategy draws inspiration from centralized methods like DKM [1], its design goals and implementation are significantly different:
>
> 1. **Different Objectives**:
>    - DKM focuses on centralized neural network compression, aiming to balance model size and task performance.
>    - AdFedWCP’s objective is to optimize communication efficiency in federated learning, addressing bandwidth heterogeneity while maintaining personalized model performance.
>
> 2. **Distinct Designs**:
>    - DKM employs a gradient-optimized differentiable k-means clustering algorithm for centralized model compression.
>    - AdFedWCP’s weight clustering pruning strategy leverages the distributed nature of federated learning, using dynamic adjustment strategies to adapt to client heterogeneity.
>
> 3. **Applicability in Different Scenarios**:
>    - DKM is suitable for single-server environments where model compression is centralized.
>    - AdFedWCP is designed for distributed federated learning environments, enabling adaptive handling of bandwidth and data distribution differences across clients.
>
> ---
>
> [1] Minsik Cho, Keivan Alizadeh-Vahid, Saurabh Adya, and Mohammad Rastegari. DKM: Differentiable k-means clustering layer for neural network compression. In International Conference on Learning Representations.

---

> ### Author Response · Authors · 2024-11-23
> **Response to Weakness 3**
>
> ### Discussion on Resource Consumption: Computational Overhead Analysis of AdFedWCP (Weakness 3)
>
> As mentioned previously, our method is not specifically tailored to handle all types of resource constraints, such as computational and storage heterogeneity. The core objective of AdFedWCP is to optimize communication efficiency and adapt to bandwidth heterogeneity, ensuring robust model performance in environments with limited bandwidth and significant data distribution differences. However, we acknowledge that the efficiency of federated learning methods is highly dependent on their resource requirements. Below, we present a detailed discussion and mathematical analysis of the resource consumption associated with AdFedWCP’s core strategies: weight clustering pruning, layer importance evaluation, and dynamic cluster center optimization.
>
> #### **1. Computational Overhead of Weight Clustering Pruning**
> The weight clustering pruning process in AdFedWCP is similar to K-means clustering, with the primary computational overhead occurring during the client-side clustering operations.
>
> 1. **Computational Complexity of the Clustering Algorithm**:
>    - Each pruning operation involves two main steps:
>      - **Assignment Step**: $\mathcal{O}(P \times K)$, where $P$ is the number of weight parameters, and $K$ is the number of centroids.
>      - **Update Step**: $\mathcal{O}(P)$.
>    - **Total Clustering Overhead**: For $T$ iterations of clustering across $L$ layers in a neural network, the total cost is:
>  $C_{\text{cluster}} = \mathcal{O}(T \times K \times \sum_{l=1}^{L} P^{(l)}).$
>
> 2. **Relative Client Overhead**:
>    Weight clustering pruning is performed only once after each training round. Compared to the overall model training cost, the additional computational overhead of pruning is negligible.
>
> #### **2. Computational Overhead of Layer Importance Evaluation**
>
> 1. **Efficiency of the Imprinting Method**:
>    Layer importance evaluation using the Imprinting method incurs computational costs equivalent to a single forward pass:
>
>    $C_{\text{imprinting}} = \mathcal{O}(\sum_{l=1}^{L} (n_{l-1} \times n_l + n_l)),$
>
>    where $n_{l-1}$ and $n_l$ are the numbers of neurons in the $(l-1)$-th and $l$-th layers, respectively.
>    - Unlike alternative methods requiring multiple iterations or complex optimizations, the Imprinting method has extremely low computational overhead, making it ideal for resource-constrained clients.
>
> 2. **Client Suitability**:
>    The lightweight design of Imprinting ensures that layer importance evaluation is highly efficient, imposing minimal computational or storage pressure on clients.
>
>
> #### **3. Computational Overhead of Dynamic Cluster Center Optimization**
> 1. **Server-Side Execution**:
>    Dynamic cluster center optimization is primarily executed on the server side. The server dynamically adjusts pruning strategies based on client bandwidth, data distribution, and layer importance, then distributes the optimized strategy to clients.
>
> 2. **Client Tasks**:
>    Clients only need to implement the received pruning strategies and update their local models, avoiding the burden of complex computations. This design minimizes the computational demands on clients.
>
> #### **4. Overall Overhead Analysis: Comparison of Training and Pruning Costs**
> Through mathematical analysis, the computational overhead of weight clustering pruning is shown to be minimal compared to the model training process.
>
> 1. **Training Cost**:
>    The total cost for forward propagation, backpropagation, and parameter updates is:
>
>    $C_{\text{train}} = \mathcal{O}(N \times E \times 3 \times (\text{total parameters})),$
>
>    where $N$ is the number of samples and $E$ is the number of local training epochs.
>
> 2. **Clustering Cost**:
>    The total cost for clustering across all layers is:
>
>    $C_{\text{cluster}} = \mathcal{O}(T \times K \times (\text{total parameters})),$
>
>    where $T$ is the number of clustering iterations, and $K$ is the number of centroids.
>
> 3. **Cost Comparison**:
>    Assuming $N = 2174$ (small client data size), $E = 1$, $T = 10$ (clustering iterations), and $K = 32$ (centroids):
>
>    $\text{Cost Ratio} = \frac{C_{\text{train}}}{C_{\text{cluster}}} = \frac{N \times E \times 3}{T \times K} = \frac{2174 \times 1 \times 3}{10 \times 32} \approx 20.38.$
>
>    - The training cost is approximately **20 times** the clustering cost, demonstrating that the computational cost of weight clustering pruning is minimal.

---

> ### Author Response · Authors · 2024-11-23
> **Response to Weakness 4 (Part A)**
>
> ###  Resource Consumption Analysis of Weight Clustering Pruning, Layer Importance Estimation and Dynamic Centroid Optimization Strategies (Weakness 4):
>
> As mentioned previously, our method is not specifically tailored to handle all types of resource constraints, such as computational and storage heterogeneity. The core objective of AdFedWCP is to optimize communication efficiency and adapt to bandwidth heterogeneity, ensuring robust model performance in environments with limited bandwidth and significant data distribution differences.  To further support this objective, we conducted ablation studies to analyze the resource consumption of key components—weight clustering pruning, layer importance estimation, and dynamic centroid optimization strategies. These analyses focus primarily on bandwidth consumption, as bandwidth heterogeneity is a central challenge in the scenarios addressed by AdFedWCP.
>
> ### Ablation Studies for Communication Overhead Consumption
> 1. **Weight Clustering Pruning**:
>    - **Ablation Study Results**: As shown in Tables 3 and 4, the variant FedWCP (w/o WCP), which does not employ weight clustering pruning, shows the highest model performance but does not optimize resource consumption, resulting in a 0% reduction in communication overhead. This highlights the efficiency of weight clustering pruning in reducing bandwidth consumption, as it significantly decreases the volume of data that needs to be communicated.
>
> 2. **Layer Importance Estimation**:
>    - **Ablation Study Analysis**: The results and analysis for this component are detailed in Appendix E.1.2. Comparing AdFedWCP with and without the imprinting method (layer importance estimation):
>      - AdFedWCP w/o imprinting on EMNIST: 85.02% accuracy, 87.29% compression rate.
>      - AdFedWCP on EMNIST: 85.12% accuracy, 87.54% compression rate.
>      - **Observations**: Both accuracy and compression rate decrease slightly without layer importance estimation, indicating its effectiveness in preserving essential model features while optimizing bandwidth use.
>
> 3. **Dynamic Centroid Optimization Strategies**:
>    - **Considerations**: This strategy does not require an ablation study per se because without it, the approach would revert to a static number of clusters akin to our FedWCP, which is already analyzed in Appendix E.1.1. Additionally, since dynamic centroid optimization is conducted server-side, it doesn’t impose resource constraints on client devices, typically a primary concern in federated learning scenarios focused on client-side computation and storage limitations.

---

> ### Author Response · Authors · 2024-11-23
> **Response to Weakness 4 (Part B)**
>
> ### Computational Overhead Analysis
>  Although computational overhead is not the focus of our research, we conducted a theoretical analysis for computational overhead. Theoretical analysis can ensure the findings are broadly applicable across various scenarios and datasets.
>  Empirical results, while valuable for specific cases, can be heavily influenced by hardware configurations and implementation details. Theoretical analysis provides a more robust, generalizable framework for understanding the computational trade-offs of weight clustering pruning, making it a suitable choice for this study.
>
> ### 1. Computational Cost of Neural Network Training
>
> #### 1.1 Neural Network Architecture
> We consider a fully connected feedforward neural network with $L$ ayers (excluding the input layer). Let layer $l$ have $n_l$ neurons, with the input layer having $n_0$ neurons.
>
> #### 1.2 Forward Propagation Cost
>
> For a single sample, forward propagation requires computing the output of each layer through:
>
> - **Linear transformation (matrix multiplication)**:
>
>   $z^{(l)} = W^{(l)} x^{(l-1)} + b^{(l)}$
>
>   Complexity:  $\mathcal O(n_{l-1} \times n_l)$
>
> - **Activation function**:
>
>   $x^{(l)} = \sigma(z^{(l)})$
>
>   Complexity: $\mathcal O(n_l)$
>
> **Single Layer Forward Propagation Cost**:
>
>   $C_{\text{forward}}^{(l)} = \mathcal O(n_{l-1} \times n_l + n_l)$
>
> **Total Forward Propagation Cost (Single Sample)**:
>
>   $C_{\text{forward}} = \sum_{l=1}^{L} C_{\text{forward}}^{(l)} = \mathcal O\left( \sum_{l=1}^{L} (n_{l-1} \times n_l + n_l) \right)$
>
>
> #### 1.3 Backward Propagation Cost
> Backward propagation involves gradient computations:
>
> - **Error term computation**:
>
>   Complexity: $\mathcal O(n_l \times n_{l+1})$
>
> - **Gradient computation**:
>
>   $\frac{\partial L}{\partial W^{(l)}} = \delta^{(l)} (x^{(l-1)})^T$
>
>   Complexity: $\mathcal O(n_{l-1} \times n_l)$
>
> **Single Layer Backward Propagation Cost**:
>
>   $C_{\text{backward}}^{(l)} = \mathcal O(n_l \times n_{l+1} + n_{l-1} \times n_l)$
>
> **Total Backward Propagation Cost (Single Sample)**:
>
>   $C_{\text{backward}} = \sum_{l=1}^{L} C_{\text{backward}}^{(l)} = O\left( \sum_{l=1}^{L} (n_l \times n_{l+1} + n_{l-1} \times n_l) \right)$
>
> #### 1.4 Parameter Update Cost
> For each layer:
> - **Weight update**:
>
>   $W^{(l)} = W^{(l)} - \eta \frac{\partial L}{\partial W^{(l)}}$
>
>   Complexity:  $\mathcal O(n_{l-1} \times n_l)$
>
> - **Bias update**:
>
>   $b^{(l)} = b^{(l)} - \eta \frac{\partial L}{\partial b^{(l)}}$
>
>   Complexity: $\mathcal O(n_l)$
>
> **Single Layer Parameter Update Cost**:
>
>   $C_{\text{update}}^{(l)} = \mathcal O(n_{l-1} \times n_l + n_l)$
>
>
> **Total Parameter Update Cost (Single Sample)**:
>
>   $C_{\text{update}} = \sum_{l=1}^{L} C_{\text{update}}^{(l)} = \mathcal O\left( \sum_{l=1}^{L} (n_{l-1} \times n_l + n_l) \right)$
>
> #### 1.5 Total Training Cost
> **Total Training Cost (Single Sample)**:
>
> $
> C_{\text{train}} = C_{\text{forward}} + C_{\text{backward}} + C_{\text{update}} = \mathcal O\left( 2 \sum_{l=1}^{L} (n_{l-1} \times n_l + n_l) + \sum_{l=1}^{L} n_l \times n_{l+1} \right)
> $
>
> **Total Training Cost for Dataset**:
>
> $C_{\text{total train}} = N \times E \times C_{\text{train}}$
>
>
> where $N$ is the number of samples in the dataset and $E$ is the number of training epochs.
>
>
> ### 2. Computational Cost of Weight Clustering
> #### 2.1 Parameter Count
> For layer$l$, the weight matrix$W^{(l)}$has:
>
>   $P^{(l)} = n_{l-1} \times n_l$
>
> #### 2.2 Clustering Algorithm Cost (e.g.,$K$-means)
>
> The $K$-means algorithm involves two main steps per iteration:
>
> - **Assignment step**:
>   Complexity: $\mathcal O(P^{(l)} \times K)$
>
> - **Update step**:
>   Complexity: $\mathcal O(P^{(l)})$
>
> **Total Clustering Cost per Layer**:
>
>   $C_{\text{cluster}}^{(l)} = T \times \mathcal O(P^{(l)} \times K + P^{(l)}) = \mathcal O(T \times P^{(l)} \times K)$
>
>   where $T$ is the number of clustering iterations.
>
>
> **Total Clustering Cost for All Layers**:
>
>   $C_{\text{total cluster}} = \sum_{l=1}^{L} C_{\text{cluster}}^{(l)} = O\left( T \times K \times \sum_{l=1}^{L} P^{(l)} \right)$
>
>
> #### 3. Training vs. Clustering Cost Ratio
>
> Using the given assumptions:
>
> - $E = 1$(local clients perform one local epoch)
>
> - $N = 2174$(the smallest client dataset size)
>
> - $K = 32$(maximum number of centroids in ablation studies)
>
> - $T = 10$(maximum clustering iteration count)
>
> ##### Training Cost:
>   $C_{\text{total train}} = N \times E \times 3 \times (\text{total parameters})$
>
> ##### Clustering Cost:
>
>   $C_{\text{total cluster}} = T \times K \times (\text{total parameters})$
>
> ##### Cost Ratio:
>
>   $\text{Cost Ratio} = \frac{C_{\text{total train}}}{C_{\text{total cluster}}} = \frac{N \times E \times 3}{T \times K}$
>
>
> ##### Substituting Values:
>
>   $\text{Cost Ratio} = \frac{2174 \times 1 \times 3}{10 \times 32} = \frac{6522}{320} \approx 20.38$
>
> This indicates that the training cost is approximately **20 times higher** than the clustering cost, emphasizing that weight clustering pruning introduces minimal computational overhead relative to training.

---

> ### Author Response · Authors · 2024-11-23
> **Response to Questions 1 and 2**
>
> ### On Whether AdFedWCP is Applicable to Heterogeneous Clients with Varying Resource Capacities (Question 1)
>
> Please reference our reponse to Weakness 1.
>
> ---
>
> ### AdFedWCP's Strategies for Resource-Constrained Clients (Question 2)
>
> As mentioned previously, our method is not specifically tailored to handle all types of resource constraints, such as computational and storage heterogeneity. The core objective of AdFedWCP is to optimize communication efficiency and adapt to bandwidth heterogeneity, ensuring robust model performance in environments with limited bandwidth and significant data distribution differences.
>
> However, while AdFedWCP is not explicitly designed for resource-constrained scenarios, its computational and storage overhead is relatively low, making it compatible with such environments to a certain extent. Below, we discuss how its core modules maintain efficiency and applicability in resource-limited settings:
>
> #### **1. Weight Clustering Pruning**
> - **Design Objective**:
>   Weight Clustering Pruning is a key module of AdFedWCP designed to reduce the number of model parameters by clustering weight values, thereby lowering communication overhead.
>
> - **Computational Overhead Analysis**:
>   This process is similar to K-means clustering, with relatively low computational complexity. It requires only a single pruning operation on the client side. Compared to more complex optimization algorithms, its computational demands are manageable, even for resource-constrained clients.
>
> - **Resource Efficiency**:
>   The pruned model significantly reduces the parameter count, which not only decreases communication costs but also reduces the computational workload for local model training on clients.
>
> #### **2. Layer Importance Evaluation**
> - **Method Selection**:
>   The Imprinting method is employed to assess layer importance. It requires only a single forward pass to complete the evaluation, offering extremely low computational overhead compared to more complex methods requiring multiple gradient calculations.
>
> - **Applicability**:
>   The efficiency of the Imprinting method makes it particularly suitable for resource-constrained clients, while still accurately evaluating layer importance to guide subsequent pruning strategies.
>
> #### **3. Dynamic Cluster Center Optimization**
> - **Execution Location**:
>   The dynamic adjustment of cluster centers is primarily performed on the server side. Clients only need to receive the clustering strategy from the server and implement the corresponding pruning and model update operations.
>
> - **Client Workload**:
>   Clients are not involved in the complex cluster optimization calculations and are only responsible for training the pruned model and updating weights. This minimizes the computational and storage demands on clients, making it suitable for resource-constrained environments.
> These design considerations ensure that AdFedWCP can operate efficiently even in scenarios with limited client resources.
>
> ---

---

### Official Review · Reviewer_SYgz · 2024-10-31

**Soundness:** 2
**Presentation:** 2
**Contribution:** 2
**Rating:** 3
**Confidence:** 4

**Summary:**

This paper presents a personalized federated learning method called AdFedWCP, which optimizes communication efficiency in heterogeneous networks through adaptive weight clustering pruning. The experimental results indicate that this method outperforms existing baseline methods in terms of both communication efficiency and accuracy across multiple datasets.

**Strengths:**

1)Innovation: The introduction of a dynamic weight clustering pruning mechanism effectively addresses issues related to bandwidth and data heterogeneity.
2)Theoretical Analysis: The paper provides detailed theoretical support, ensuring the validity of the proposed method.
3)Comprehensive Experimental Design: The experiments cover multiple datasets, validating the performance of the method.

**Weaknesses:**

1)Insufficient Support for Cluster Number Selection: While the idea of dynamically adjusting the number of clusters is novel, the paper lacks detailed theoretical justification and empirical data to support the selection criteria for the number of clusters. It is recommended to include relevant discussions.
2)Inadequate Analysis of Experimental Results: The interpretation of results is somewhat superficial and fails to explore the reasons for performance deficiencies on specific datasets (e.g., CIFAR-100). Particularly, the accuracy on CIFAR-10 and CIFAR-100 does not reach optimal levels, indicating that the method's performance on more complex datasets requires improvement.
3)Comparison with FedWCP on K Values: In the comparison with different K values of FedWCP, AdFedWCP did not consistently outperform other configurations, suggesting that the balance between model complexity and performance remains an issue.
4)Lack of Comparative Experiments: There is insufficient comparison with the latest related methods, such as [1], resulting in an inadequate assessment of AdFedWCP's effectiveness. Including comparisons with other advanced methods would strengthen the conclusions.
5)Inadequate Discussion of Limitations: The conclusion section does not address the limitations of AdFedWCP. It is recommended to analyze its performance in extremely heterogeneous environments and to provide suggestions for future research directions.
6)Insufficient Discussion on the Trade-off between Communication Efficiency and Model Performance: While the experimental results show improvements in communication efficiency, there is a lack of in-depth discussion on the trade-off between communication costs and model complexity, which may lead to insufficient understanding of the method's applicability.
7)Incomplete Ablation Study: The paper fails to adequately discuss the effectiveness of different layer importance evaluation strategies, particularly the comparison between the imprinting method and other evaluation methods (e.g. [2]). This is crucial for optimizing model compression strategies.
[1] Wu C, Wu F, Lyu L, et al. Communication-efficient federated learning via knowledge distillation[J]. Nature communications, 2022, 13(1): 2032.
[2] Ma X, Zhang J, Guo S, et al. Layer-wised model aggregation for personalized federated learning[C]//Proceedings of the IEEE/CVF conference on computer vision and pattern recognition. 2022: 10092-10101.

**Questions:**

See the weakness.

---

> ### Author Response · Authors · 2024-11-23
> **Response to Weakness 1**
>
> ### Regarding the Lack of Theoretical Support for Dynamic Cluster Number Selection (Weakness 1):
> We appreciate the reviewer’s valuable feedback on the need for specific theoretical justification for each factor considered in the dynamic cluster number selection process. Below, we provide a detailed response, elaborating on the theoretical basis and empirical validation for each criterion in our methodology:
>
> #### **1. Discussion of Cluster Number Selection Criteria in the Paper**
> Our approach considers several dynamic factors influencing the selection of cluster numbers. Each factor is supported by theoretical insights and validated by experimental results:
>
> - **Data Volume**:
>   - **Theoretical Basis**: Clients with larger data volumes provide richer feature representations, requiring more centroids to capture subtle patterns effectively. This is supported by prior research on data complexity and feature representation [1] [2].
>   - **Empirical Validation**: The empirical validation of our theoretical basis regarding data volume is demonstrated in the experiments conducted in study [1].
>
> - **Bandwidth**:
>   - **Theoretical Basis**: For clients with limited bandwidth, reducing their communication costs is essential [3]. And this can be effectively achieved by reducing the number of centroids.
>   - **Empirical Validation**: As shown in the ablation study (Appendix E.1.1), reducing the cluster number significantly enhances communication efficiency. Additionally, our experiments in Appendix E.7 demonstrate that lower-bandwidth clients achieve acceptable performance even with reduced cluster numbers, highlighting the trade-off between bandwidth constraints and communication efficiency.
>
>
>
> - **Layer Importance**:
>   - **Theoretical Basis**: Neural network layers contribute unequally to the overall model. Prioritizing more centroids for important layers ensures that key features are preserved [4].
>   - **Empirical Validation**: As demonstrated in Appendix E.1.2, omitting layer importance evaluation leads to a decline in both model performance and communication compression rates compared to the full method, underlining the necessity of considering this factor.
>
>
>
> - **Training Progress and Accuracy Changes**:
>   - **Theoretical Basis**: In the later stages of neural network training, the model requires more precise representations to capture finer details and improve convergence [5]. Therefore, the representation of the model is enhanced by increasing the number of centroids.
>   - **Empirical Validation**: Our ablation studies (Appendix E.1.1) show that gradually increasing the number of centroids during training improves overall model performance, confirming the utility of this dynamic adjustment.
>
> #### **2. Integration of Theoretical Concepts**
> We have adapted established clustering, optimization, and federated learning theories to address the unique challenges posed by data and bandwidth heterogeneity in federated environments. This theoretical integration ensures that:
> - Each factor considered in cluster number selection is grounded in related research.
> - The overall dynamic cluster selection strategy is tailored to balance communication efficiency and model performance in practical scenarios.
>
> ---
>
> [1] Chen Sun, Abhinav Shrivastava, Saurabh Singh, and Abhinav Gupta. Revisiting unreasonable effectiveness of data in deep learning era. In Proceedings of the IEEE international conference on computer vision, pp. 843–852, 2017
>
> [2] Connor Shorten and Taghi M Khoshgoftaar. A survey on image data augmentation for deep learning. Journal of big data, 6(1):1–48, 2019
>
> [3] Wei Yang Bryan Lim, Nguyen Cong Luong, Dinh Thai Hoang, Yutao Jiao, Ying-Chang Liang, Qiang Yang, Dusit Niyato, and Chunyan Miao. Federated learning in mobile edge networks: A comprehensive survey. IEEE Communications Surveys Tutorials, 22(3):2031–2063, 2020. doi: 10.1109/COMST.2020.2986024.
>
> [4] Hongyang Liu, Sara Elkerdawy, Nilanjan Ray, and Mostafa Elhoushi. Layer importance estimation with imprinting for neural network quantization. In Proceedings of the IEEE/CVF Conference on Computer Vision and Pattern Recognition, pp. 2408–2417, 2021.
>
> [5] Nasim Rahaman, Aristide Baratin, Devansh Arpit, Felix Draxler, Min Lin, Fred Hamprecht, Yoshua Bengio, and Aaron Courville. On the spectral bias of neural networks. In International conference on machine learning, pp. 5301–5310. PMLR, 2019.

---

> ### Author Response · Authors · 2024-11-23
> **Response to Weakness 2**
>
> ### Further Explanation on Experimental Analysis and Performance Issues on CIFAR-10 and CIFAR-100 (Weakness 2)
>
> We appreciate the reviewer’s feedback regarding the need for a deeper analysis of experimental results. Below, we provide detailed explanations and analyses of the performance issues observed on the CIFAR-10 and CIFAR-100 datasets.
>
>
> #### **1. Reasons for Lower Performance on CIFAR-100**
> In our experiments, we employed the relatively simple LeNet neural network model for evaluation. This choice was primarily driven by the need for fair comparisons with baseline methods, such as pFedGate:
> - **Model Complexity Limitation**: Methods like pFedGate face challenges when scaling to more complex architectures (e.g., ResNet) [6]. To ensure the comparability and completeness of the experimental results, we opted for LeNet used in pFedGate as a unified model architecture.
> - **Impact**: This choice constrained the method’s performance on more complex datasets like CIFAR-100, which features higher complexity (more classes and more intricate feature distributions) and demands deeper models to fully exploit the data characteristics.
>
>
> #### **2. Reasons for Suboptimal Accuracy on CIFAR-10 and CIFAR-100**
> When compared to certain baseline methods (e.g., FedEM), the performance of AdFedWCP on CIFAR-10 and CIFAR-100 was slightly lower. The reasons are as follows:
> 1. **Characteristics of Baseline Methods**:
>    - **FedEM’s Optimization Focus**: FedEM is tailored for personalized federated learning (PFL) and is designed to maximize the performance of personalized models without considering communication costs or bandwidth heterogeneity. Under ideal conditions (i.e., no communication constraints), this allows FedEM to achieve higher accuracy levels.
>    - **AdFedWCP’s Optimization Focus**: Our method primarily addresses bandwidth heterogeneity and communication cost issues while also considering model personalization. Under more constrained communication and resource conditions, AdFedWCP significantly reduces communication overhead while maintaining accuracy close to or surpassing FedEM, showcasing its practical value.
>
> 2. **Dataset Complexity**:
>    - The complexity differences between CIFAR-10 and CIFAR-100 also impacted performance:
>    - On CIFAR-10 (a simpler dataset), AdFedWCP achieved near-optimal performance.
>    - On CIFAR-100 (a more complex dataset), the simplicity of the LeNet model, combined with the communication-focused optimization goals, resulted in slightly reduced performance.
>
> #### **3. Advantages of AdFedWCP**
>
> Despite slightly lower performance in some cases compared to FedEM, AdFedWCP demonstrates significant overall advantages:
> - **Communication Efficiency**: Through the dynamic weight clustering and pruning mechanism, AdFedWCP substantially reduces communication costs, which is critical in real-world applications.
> - **Performance Stability**: Under bandwidth heterogeneity and communication constraints, AdFedWCP achieves stable performance, close to the optimal levels of baseline methods.
> - **Practical Applicability**: AdFedWCP is better suited for resource-constrained environments, maintaining high model performance while addressing real-world challenges effectively.
>
> ---
>
> [6] Daoyuan Chen, Liuyi Yao, Dawei Gao, Bolin Ding, and Yaliang Li. Efficient personalized federated learning via sparse model-adaptation. In International Conference on Machine Learning, pp. 5234–5256. PMLR, 2023

---

> ### Author Response · Authors · 2024-11-23
> **Response to Weakness 3**
>
> ### Analysis of AdFedWCP Performance Under Different $K$ Values (Weakness 3)
>
> We appreciate the reviewer's detailed inquiry into this matter. Below, we provide further analysis and explanation of the performance of AdFedWCP and FedWCP under different $K$ configurations:
>
> #### **1. Balancing Model Complexity and Performance**
> As noted, experimental results show that AdFedWCP does not outperform all fixed $K$ configurations of FedWCP in terms of performance. However, AdFedWCP's key advantage lies in its ability to achieve a **better balance between communication cost and model performance**. Supporting evidence from the experiments includes:
> - **Performance Comparison**:
>   From Table 3 in the paper, AdFedWCP achieves accuracies of **61.04%** and **20.44%** on CIFAR-10 and CIFAR-100, respectively, which are close to the performance of FedWCP with a fixed $K=32$ (**63.38%** and **20.96%**). At the same time, AdFedWCP significantly outperforms configurations with $K=8$ and $K=16$.
>
> - **Communication Cost Comparison**:
>   From Table 4, the communication cost of AdFedWCP (87.82% on CIFAR-10 and 87.54% on CIFAR-100) is lower than that of FedWCP with $K=32$ and is close to the efficiency of the $K=16$ configuration (87.41% and 87.36%, respectively).
>
> These results indicate that AdFedWCP strikes a superior balance between performance and communication cost. This dynamic adjustment mechanism is particularly advantageous in scenarios with significant bandwidth heterogeneity.
>
> #### **2. Advantages of Dynamic Adjustment**
> - **Eliminates Manual Tuning**:
>   Fixed-$K$ FedWCP requires manual selection of suitable parameters, which can be time- and resource-intensive. AdFedWCP's dynamic adjustment mechanism automatically adaptsthe $K$ value based on client bandwidth, data distribution, and layer importance, eliminating the complexity of manual tuning. This adaptability is particularly important in real-world applications.
>
> - **Balancing Performance and Communication Efficiency**:
>   AdFedWCP's dynamic adjustment mechanism ensures **robustness across varying bandwidth and data heterogeneity conditions**, unlike fixed-$K$ FedWCP, which cannot accommodate all scenarios. For example:
>   - Under sufficient bandwidth, AdFedWCP tends to use larger cluster numbers to enhance model performance.
>   - Under constrained bandwidth, AdFedWCP reduces the cluster number automatically, minimizing communication overhead.
>
>
> #### **3. Limitations of Fixed $K$ Configurations**
>
> While FedWCP with $K=32$ achieves slightly higher performance than AdFedWCP in some scenarios, this advantage primarily arises from the greater adaptability of a larger cluster number to complex datasets. However, this configuration comes with significantly increased communication costs, making it less practical. In contrast:
> - **Adaptability of AdFedWCP**: Under equivalent communication cost conditions, AdFedWCP maintains near-optimal performance.
> - **Resource Balancing**: AdFedWCP's dynamic mechanism provides a "middle ground" solution, offering an effective balance between performance and communication efficiency, even if its performance is slightly lower than the best fixed-$K$ configuration.
>
> #### **4. Non-Adaptive Variant of AdFedWCP**
>
> To further clarify, it is worth noting that the core mechanism of AdFedWCP lies in its dynamic adjustment of $K$. When this dynamic adjustment is disabled, AdFedWCP effectively reduces to a fixed-$K$ FedWCP configuration. Thus, AdFedWCP can function both as a dynamic adjustment method and as a fixed-$K$ method.
>
> This flexibility underscores the practicality and adaptability of AdFedWCP. For scenarios where time and cost are not concerns, manually tuned fixed-$K$ methods might achieve optimal performance under specific conditions. However, in real-world applications, AdFedWCP's adaptive nature provides a more efficient and practical solution.
>
> ---

---

> ### Author Response · Authors · 2024-11-23
> **Response to Weakness 4**
>
> ### Regarding the Lack of Comparative Experiments with Recent Methods (Weakness 4)
>
> We appreciate the reviewer’s valuable suggestion. We understand that comparisons with related methods (e.g., FedKD [7]) could further strengthen the evaluation of AdFedWCP's effectiveness. However, we would like to clarify that FedKD and AdFedWCP differ significantly in their objectives and application scenarios, which might limit the fairness of direct comparisons. Below, we provide our detailed response and supplementary experimental analysis:
>
> #### **1. Differences in Research Objectives Between AdFedWCP and FedKD**
>
> - **FedKD’s Objectives and Scenarios**:
>   FedKD primarily aims to reduce communication costs through knowledge distillation, specifically designed for scenarios involving **heterogeneous model architectures across clients**. It addresses challenges related to communication and collaboration when clients use different model architectures.
>
> - **AdFedWCP’s Objectives and Scenarios**:
>   AdFedWCP, on the other hand, focuses on tackling **data and bandwidth heterogeneity** by leveraging a dynamic weight clustering and pruning mechanism. While ensuring high model performance, AdFedWCP significantly reduces communication costs, assuming consistent model architectures across clients but with diverse bandwidth and data distributions.
>
> Thus, the two methods target different problems under different assumptions, which might impact the fairness of direct comparisons.
>
> #### **2. Experimental Setup Explanation**
>
> Despite the differences in objectives, we designed and conducted supplementary experiments to compare FedKD and AdFedWCP, investigating the performance of knowledge distillation versus dynamic pruning under identical conditions. To ensure fairness in the comparison, we made the following adjustments:
>
> 1. **Homogeneous Model Architectures Across Clients**:
>    To avoid the influence of heterogeneous models, we configured all clients to use the same model architecture (consistent with AdFedWCP).
>
> 2. **Consistency in Shared Models**:
>    In FedKD’s global knowledge distillation, the knowledge shared among clients was derived from models trained using identical architectures, ensuring fairness in comparison.
>
> #### **3. Experimental Results and Analysis**
>
> | Method      | Accuracy | Compression Rate |
> |-------------|----------|------------------|
> | FedKD       | 84.06%   | 73.176%          |
> | AdFedWCP    | **85.12%** | **87.54%**       |
>
> - **Accuracy Comparison**:
>   On the EMNIST dataset, AdFedWCP achieved higher accuracy than FedKD (85.12% vs. 84.06%), indicating that dynamic weight clustering and pruning better accommodate data heterogeneity, enhancing model performance.
>
> - **Compression Rate Comparison**:
>   AdFedWCP significantly outperformed FedKD in communication compression (87.54% vs. 73.176%), demonstrating the superiority of dynamic pruning in reducing communication overhead.
>
> **Analysis of Results**:
>
> 1. **Differences in Performance**:
>    While FedKD compresses communication through knowledge distillation, it is not explicitly designed to optimize for bandwidth heterogeneity, which could limit its efficiency. In bandwidth-constrained environments, AdFedWCP's dynamic adjustment mechanism is better suited.
>
> 2. **Ensuring Fairness**:
>    If heterogeneous models (FedKD’s primary scenario) were used, the results could reflect inconsistent model performance, potentially compromising fairness. Thus, we conducted the comparison under homogeneous model architectures to ensure controlled and fair evaluations.
>
> These experiments demonstrate that AdFedWCP is more effective in balancing performance and communication efficiency under bandwidth and data heterogeneity, while maintaining fairness in the comparative analysis.
>
> ---
>
> [7] Chuhan Wu, Fangzhao Wu, Lingjuan Lyu, Yongfeng Huang, and Xing Xie. Communication-efficient federated learning via knowledge distillation. Nature communications, 13(1):2032, 2022

---

> ### Author Response · Authors · 2024-11-23
> **Response to Weakness 5**
>
> ### Regarding the Insufficient Discussion of AdFedWCP's Limitations (Weakness 5)
>
> We appreciate the reviewer's suggestion. We acknowledge that analyzing the limitations of our method is a crucial component of a comprehensive evaluation of its applicability. A detailed discussion on the extreme heterogeneity addressed by AdFedWCP is presented in Appendix E.7 (see below as well), while its potential limitations are outlined in Appendix F (see below as well).
>
> ### Supplementary Experiment: Performance Evaluation Under Extreme Heterogeneity
> To further assess AdFedWCP's adaptability in extreme heterogeneous scenarios, we designed and conducted the following experiments:
> - **Experimental Setup**:
>    Clients were divided into five distinct bandwidth groups (e.g., low-bandwidth group, medium-bandwidth group, and high-bandwidth group) to evaluate AdFedWCP's performance under varying bandwidth conditions.
>
> - **Results**:
> | Bandwidth Range       | Average Accuracy (%) |
> |------------------------|----------------------|
> | 5 Mbps - 24 Mbps       | 86.55%               |
> | 24 Mbps - 43 Mbps      | 86.41%               |
> | 43 Mbps - 62 Mbps      | 85.08%               |
> | 62 Mbps - 81 Mbps      | 85.87%               |
> | 81 Mbps - 100 Mbps     | 85.15%               |
>
> The experimental results indicate that **the accuracy of AdFedWCP varies minimally across different bandwidth groups** (differences of less than 1.5%). This demonstrates:
> 1. **Performance Stability**:
>   Even in the lowest bandwidth group (5 Mbps - 24 Mbps), AdFedWCP achieves an average accuracy of **86.55%**, indicating that the dynamic weight clustering and pruning mechanism effectively preserves model performance across diverse bandwidth conditions.
>
> 2. **Adaptability of the Dynamic Pruning Mechanism**:
>   AdFedWCP's dynamic adjustment strategy ensures that pruning rates and model complexity adapt to the actual environment, effectively balancing communication efficiency and model performance.
>
> ### POTENTIAL LIMITATIONS OF ADFEDWCP
>
> While AdFedWCP demonstrates strong performance in addressing bandwidth heterogeneity and improving communication efficiency, there are several limitations that warrant consideration for future research and practical applications. First, AdFedWCP assumes static communication bandwidths for clients throughout the training process. While this simplifies the experimental setup and enables controlled evaluation, real-world federated learning systems often encounter dynamic bandwidth fluctuations. Incorporating mechanisms to address dynamic bandwidth changes could further enhance the robustness and adaptability of AdFedWCP in practical deployments.
>
> Second, AdFedWCP uses the Imprinting method for layer importance evaluation, which is computationally efficient and well-suited for resource-constrained federated learning environments. However, this approach limits the evaluation to a single method, leaving the potential benefits of other importance evaluation techniques unexplored. Future work could investigate alternative methods, such as saliency-based or gradient-based techniques, to optimize the pruning strategy further and improve model performance.
>
> Lastly, AdFedWCP primarily focuses on addressing bandwidth heterogeneity among clients, with limited consideration for computational heterogeneity, such as differences in processing power or memory capacity across devices. While the weight clustering pruning mechanism has the potential to reduce computational overhead by creating sparse matrices, the sparsity generated by clustering may not be structured. As a result, the improvement with computational efficiency is not as significant as that of structured sparsity methods. Extending AdFedWCP to explicitly address computational heterogeneity or structured sparsity could significantly enhance its applicability in highly resource-constrained environments.
>
> These limitations suggest clear directions for future work, including the development of strategies to handle dynamic bandwidth, exploration of alternative layer importance evaluation methods, and explicit optimization for computational heterogeneity. Addressing these challenges could further enhance the adaptability, scalability, and efficiency of AdFedWCP in diverse and practical federated learning scenarios.

---

> ### Author Response · Authors · 2024-11-23
> **Response to Weakness 6 (Part A)**
>
> ### Discussion on the Trade-Off Between Communication Efficiency and Model Performance (Weakness 6)
> We fully agree that the trade-off between communication efficiency and model complexity is a critical issue in federated learning. Below is a detailed explanation of how AdFedWCP addresses this trade-off, supported by experimental evidence.
>
> #### **1. Model Complexity and Compression Rates**
> Experimental data from Table 2 and Table 7 in the paper demonstrates that AdFedWCP achieves similar compression rates across different model architectures, despite variations in model complexity:
> - For LeafCNN (EMNIST), AdFedWCP achieves a compression rate of **87.54%**.
> - For LeNet (CIFAR-10), the compression rate is **87.82%**.
>
> An analysis of the parameter counts for the LeafCNN and LeNet models reveals that LeafCNN is significantly more complex than LeNet. Designed for the EMNIST dataset, the LeafCNN model comprises approximately 2,227,590 trainable parameters, featuring multiple convolutional layers and a much larger fully connected layer configuration. In contrast, LeNet, commonly used for CIFAR-10, has a simpler architecture with around 256,830 parameters, including fewer convolutional layers and smaller fully connected layers.
>
> This stark contrast in parameter counts, with LeafCNN having approximately 8.67 times more parameters than LeNet, indicates a much higher level of complexity. This disparity arises primarily from LeafCNN's additional convolutional layers and extensive fully connected layers, which are tailored to capture more complex patterns and finer-grained details required to handle the more diverse and detailed class structures in EMNIST.
>
> Despite these differences in underlying model complexity, the compression rates achieved using AdFedWCP are similar for both models: 87.54% for LeafCNN and 87.82% for LeNet. This near-identical compression performance highlights the effectiveness of the AdFedWCP methodology, which is capable of significantly reducing communication overhead without being adversely affected by the inherent complexity of the neural network architectures. This robustness demonstrates AdFedWCP's versatility across diverse architectures, ensuring efficient communication.
>
> ##### **Mathematical Perspective**
> From the mathematical analysis of WCP, the compression rate in federated learning is primarily determined by:
> - **Number of Weights ($N$)**: The total number of model parameters.
> - **Number of Clusters ($k$)**: The number of centroids used for weight clustering.
> - **Representation Cost**: The number of bits required to transmit centroid values and the index sequence.
>
> The total communication cost for WCP can be expressed as:
> $C_{\text{WCP}} = (k - 1) \cdot B + N \cdot \lceil \log_2 k \rceil,$
>
> where $B$ is the bit-width of centroids. Notably, this cost scales linearly with the total number of weights $N$ and logarithmically with the number of clusters $k$.
>
> Importantly, for different model architectures:
> - **The parameter count ($N$)** differs across models, but the compression mechanism (clustering weights into $k$ centroids) normalizes this variation. Thus, the relative impact of model complexity on compression rates is mitigated.
> - **Cluster-based representation** abstracts individual weight differences within a model, meaning that the architecture's complexity does not strongly affect how well weights can be clustered and compressed.
> - The dominant factor in compression efficiency is the choice of $k$, not the inherent architectural complexity of the model.
>
> #### **2. Balancing Communication Efficiency and Model Performance**
> The core design of AdFedWCP achieves an effective balance between communication efficiency and model performance through dynamic weight clustering and pruning. Specifically:
> - **Ensuring Communication Efficiency**:
>   By dynamically adjusting the pruning rate and the number of cluster centroids, AdFedWCP significantly reduces communication costs.
>
> - **Stabilizing Model Performance**:
>   It simultaneously preserves important model features, minimizing the negative impact of pruning on performance.
>
> **Experimental Support**:
> - **Performance of Full AdFedWCP**:
>   - On the EMNIST dataset, AdFedWCP achieved **85.12% accuracy** with **87.54% compression**, demonstrating near-optimal performance while significantly reducing communication overhead.
>
> - **Comparison with Fixed Pruning Strategies**:
>   - As shown in Table 3 and Table 4, AdFedWCP achieved **61.04% accuracy** on the CIFAR-10 dataset, close to the fixed pruning strategy with $K=32$ (**63.38%**), while achieving significantly higher communication efficiency (compression rate **87.82%** vs. **84.21%**).
>   - These results highlight that AdFedWCP finds an excellent balance between performance and communication efficiency, particularly suitable for scenarios with significant bandwidth heterogeneity.

---

> ### Author Response · Authors · 2024-11-23
> **Response to Weakness 6 (Part B)**
>
> #### **3. Ablation Study: The Role of Upper and Lower Bounds**
>
> To further explore the trade-off between communication efficiency and model performance, we conducted ablation studies analyzing the effects of the upper and lower bounds in AdFedWCP:
>
> 1. **Without the Lower Bound**:
>    - **Results**: Accuracy dropped to **78.27%**, while compression rate increased to **90.53%** on the EMNIST dataset.
>    - **Analysis**: Without the lower bound, excessive pruning led to higher compression at the cost of degraded model performance. This demonstrates the critical role of the lower bound in maintaining model performance.
>
> 2. **Without the Upper Bound**:
>    - **Results**: Accuracy remained at **85.12%**, but compression rate decreased to **87.37%** on the EMNIST dataset..
>    - **Analysis**: Without the upper bound, insufficient pruning led to higher communication costs, while model performance was unaffected. This highlights the importance of the upper bound in controlling communication overhead.
>
> The ablation studies validate that the upper and lower bounds are crucial in balancing communication efficiency and model performance:
> - The lower bound ensures model performance.
> - The upper bound effectively limits communication costs.
>
> #### **4. The Role of Dynamic Adjustment Mechanism**
>
> AdFedWCP's dynamic adjustment mechanism flexibly modifies pruning rates based on the actual resources of clients, achieving an adaptive balance between communication efficiency and model performance under different conditions:
>
> - **Communication-Constrained Scenarios**:
>   Under low bandwidth conditions, AdFedWCP prioritizes reducing the number of centroids to minimize communication costs while preserving key model features.
>
> - **Resource-Rich Scenarios**:
>   Under high bandwidth conditions, AdFedWCP dynamically increases the number of centroids and model representation capacity to enhance model performance.
>
> This dynamic mechanism enables AdFedWCP to adapt to varying bandwidth constraints and heterogeneity while maintaining a robust balance between performance and communication efficiency.

---

> ### Author Response · Authors · 2024-11-23
> **Response to Weakness 7**
>
> ### Supplementary Ablation Studies and Discussion on Layer Importance Evaluation (Weakness 7)
>
> We appreciate the reviewer’s valuable suggestions. Below, we provide additional details on the use of the Imprinting method for layer importance evaluation and supplementary experiments to support its effectiveness.
>
> #### **1. Use of the Imprinting Method and Rationale for Selection**
> We emphasize that the Imprinting method is not presented as a novel contribution but as an effective tool integrated into our framework for evaluating layer importance. The reasons for choosing the Imprinting method are as follows:
> - **Efficiency**:
>   The Imprinting method requires only a single forward pass to evaluate layer importance, making it particularly suitable for resource-constrained federated learning scenarios by significantly reducing computational and communication costs.
>
> - **Applicability**:
>   The Imprinting method has been demonstrated in prior studies (e.g.,[5]) to perform well in optimizing model compression strategies. We integrated it into our dynamic weight clustering and pruning strategy to leverage its proven effectiveness.
>
> By incorporating the Imprinting method into AdFedWCP, we can efficiently evaluate layer importance, which aids in balancing communication efficiency and model performance.
>
> #### **2. Supplementary Ablation Experiments**
> To validate the effectiveness of layer importance evaluation using the Imprinting method, we conducted a comparative experiment between:
> - AdFedWCP with layer importance evaluation (using the Imprinting method);
> - AdFedWCP without layer importance evaluation.
> **Results**:
>
> | Method                      | EMNIST Accuracy  | Compression Rate |
> |---------------------------|-----------|------------------|
> | AdFedWCP w/o imprinting      | 85.02%    | 87.29%           |
> | AdFedWCP                     | **85.12%** | **87.54%**        |
>
> **Analysis**:
> - **Performance Improvement**:
>   AdFedWCP with layer importance evaluation achieved slightly higher accuracy than the version without it, indicating that layer importance evaluation effectively identifies critical layers and enhances their representation, thereby improving performance.
>
> - **Increased Compression Efficiency**:
>   AdFedWCP with layer importance evaluation also achieved a higher compression rate, demonstrating that the evaluation method better guides dynamic weight clustering and pruning, reducing communication overhead while maintaining model performance.
>
>
> #### **3. Necessity of Layer Importance Evaluation**
>
> The experimental results demonstrate that layer importance evaluation plays the following roles in dynamic weight clustering and pruning:
>
> 1. **Optimizing Pruning Strategy**:
>    By evaluating the importance of each layer, the Imprinting method prioritizes retaining layers that contribute most to model performance, enabling a better balance between performance and communication efficiency during pruning.
>
> 2. **Enhancing Model Performance**:
>    Compared to methods without layer importance evaluation, AdFedWCP achieves higher accuracy while maintaining high communication efficiency.
>
> #### **4. Prospects for Alternative Evaluation Methods**
>
> While the Imprinting method demonstrates good efficiency and applicability, we recognize that exploring other layer importance evaluation methods might further improve AdFedWCP’s performance. In future work, we plan to:
> 1. Compare the advantages and disadvantages of alternative layer importance evaluation strategies.
> 2. Integrate more advanced evaluation methods into AdFedWCP to further optimize the pruning strategy.

---

### Official Review · Reviewer_61kH · 2024-11-03

**Soundness:** 2
**Presentation:** 2
**Contribution:** 2
**Rating:** 6
**Confidence:** 3

**Summary:**

This paper proposes an Adaptive Federated Weight Clustering Pruning (AdFedWCP) to address the data and communication heterogeneity issues. Specifically, an imprinting method is designed to calculate the importance level of each layer; then, the number of clustering centroids is determined by the server based on the client's data and communication information; finally, each client executes the weight clustering pruning. Experimental results based on three datasets show that the model accuracy of  AdFedWCP is boosted while the communication overhead is reduced compared to baselines.

**Strengths:**

1. There are both theoretical and experimental analyses.

2. The layer importance and weight cluster pruning are valuable research problems for heterogeneous federated learning.

**Weaknesses:**

1. More than half of the abstract's content is about the experimental result, while the experimental part of the paper is only two pages long. I suggest the abstract should summarize more key points and the main body should include more result analysis.

2. Authors claim many times that one main advantage of AdFedWCP is addressing the bandwidth heterogeneity problem that many other methods can not handle. However, there is no detailed explanation of the bandwidth heterogeneity problem and no clear definition of bandwidth heterogeneity.

3. The authors claim that their research problem is to minimize communication costs and address the bandwidth heterogeneity problem without compromising model accuracy. However, the optimization objective defined in section 3 is not related to communication costs/bandwidth heterogeneity.

4. Figure 1 should be refined to illustrate more clearly how these values are calculated and show the overall framework of  AdFedWCP.

5. Only the final results are in the experiment part, and the learning curves (e.g., communication cost vs. model accuracy) are missing.

6. The description of the client heterogeneity settings (lines 465-467) is not detailed enough. Since bandwidth heterogeneity is the key point in this paper, related experimental settings should be more carefully designed and described in more detail. For a client, will its bandwidth also change over time?

7. The limitations of AdFedWCP should also be discussed, except for the advantages.


8. For Table 1 and Table 2, the best value should be bolded, and the second best value should be underlined for better comparison and readability.

**Questions:**

1. There are several hyper-parameters in the proposed method, e.g., lines 130-131 and lines 366-367. Will different values of these hyper-parameters affect performance a lot? Why or why not? How to choose proper values for implmentations?

2. Can authors provide more learning curves (e.g., communication cost vs. model accuracy) of AdFedWCP and other baselines to show the stability and comparison?

3. How many clients will participate in the training among all clients? There is no such information.

4. For a client, will its bandwidth also change over time in the experiment?

---

> ### Author Response · Authors · 2024-11-23
> **Response to Weaknesses 1 and 2**
>
> ### Revised Abstract Overview (Weakness 1)
> We have revised the abstract in ther revised version to include a more balanced summary of the key points of our work (see below).
>
> #### Revised abstract
> This paper introduces a novel personalized federated learning approach, Adaptive Federated Weight Clustering Pruning (AdFedWCP), specifically designed to optimize communication efficiency in heterogeneous network environments. AdFedWCP innovatively combines adaptive weight clustering pruning techniques, effectively addressing data and bandwidth heterogeneity. By dynamically adjusting clustering centroids based on layer importance and client-specific data characteristics, it significantly reduces communication overhead. Experimental results demonstrate reductions in communication volume by up to 87.82\% and accuracy improvements of 9.13\% to 21.79\% over baselines on EMNIST, CIFAR-10, and CIFAR-100. These findings underscore AdFedWCP’s effectiveness in balancing communication efficiency and model accuracy, making it suitable for resource-constrained federated learning.
>
> ---
>
> ### Regarding the Definition and Discussion of Bandwidth Heterogeneity (Weakness 2)
> We appreciate the reviewer’s suggestion. We acknowledge that the definition and detailed explanation of "bandwidth heterogeneity" could be made clearer and more comprehensive in the paper. Below, we provide the definition and a summary of the related discussions in the paper:
>
> #### **1. Definition of Bandwidth Heterogeneity**
> Bandwidth heterogeneity refers to the significant variation in available communication bandwidth across clients in a federated learning system due to differences in their network environments. This phenomenon can manifest as follows:
> - Some clients operate in high-speed, stable network environments (e.g., corporate intranets or home broadband).
> - Others may be in low-speed or unstable network conditions (e.g., mobile devices relying on public Wi-Fi or cellular networks).
>
> The uneven distribution of bandwidth introduces the following challenges to federated learning systems:
> - **Communication Efficiency Bottlenecks**: Clients with limited bandwidth may become bottlenecks for global model updates, reducing overall system efficiency.
> - **Fairness Issues**: In certain existing methods, bandwidth-constrained clients may be entirely excluded, preventing their data from contributing to the global model.
>
> We defined bandwidth heterogeneity in lines 41-43 on page 1 of the introduction.
>
> #### **2. Discussion of Bandwidth Heterogeneity in the Paper**
> We explained the issue of bandwidth heterogeneity in several sections of the paper and proposed a dedicated mechanism (dynamic clustering and pruning) to mitigate its impact. Relevant details include:
> 1. **Introduction (Page 1, Lines 41-43)**:
>    In the introduction, we highlight the prevalence and challenges of bandwidth heterogeneity in real-world federated learning scenarios. We discuss how existing methods may face efficiency and fairness issues under bandwidth-constrained environments, setting the stage for the proposed AdFedWCP method.
>
> 2. **Experiments (Page 8, Lines 465-466)**:
>    In the experimental setup, we simulate bandwidth distributions ranging from **5 Mbps to 100 Mbps** to validate the effectiveness of AdFedWCP in handling bandwidth heterogeneity. The results demonstrate that the **dynamic clustering and pruning mechanism** can automatically adjust pruning rates based on client bandwidth, significantly reducing communication overhead while maintaining model performance.
>
> This expanded discussion clarifies how bandwidth heterogeneity is defined and tackled in the paper, emphasizing the practicality and robustness of AdFedWCP in addressing real-world challenges.

---

> ### Author Response · Authors · 2024-11-23
> **Response to Weaknesses 3, 4 and 5**
>
> ### Clarification on the Relation Between the Optimization Objective in Section 3 and Communication Cost (Weakness 3)
> We appreciate the reviewer’s observation regarding the potential misunderstanding about the relationship between the optimization objective in Section 3 and communication costs. Below, we provide a detailed clarification and supplementary explanation:
>
> #### **1. Core Purpose of the Optimization Objective in Section 3**
> The optimization objective defined in Section 3 is designed for personalized federated learning, with the following primary goals:
>
> - **Local Model Adaptability**: By minimizing the loss on local data, the objective ensures that models can adequately capture the unique data distribution characteristics of each client.
> - **Global Knowledge Sharing**: By introducing a global model regularization term, the objective facilitates knowledge sharing across clients to address data heterogeneity.
>
>
> **Relation to Communication Cost**:
> The optimization objective in Section 3 represents the foundational framework of federated learning, aiming to balance global and local model performance. This objective is decoupled from the optimization of communication costs, which is treated as a secondary problem discussed explicitly in Section 4.4.
>
> #### **2. Communication Cost Minimization Addressed in Section 4.4**
> Section 4.4 is dedicated to addressing the minimization of communication costs through the dynamic clustering and pruning strategy. This section focuses on:
> - **Dynamic Pruning Mechanism**: Adjusting the weight pruning rates for each client based on their bandwidth distribution to significantly reduce communication overhead.
> - **Bandwidth Heterogeneity Adaptation**: Dynamically adjusting pruning rates and the number of cluster centers to ensure efficient participation of clients with varying bandwidths, preventing low-bandwidth clients from becoming communication bottlenecks.
>
> The separation of communication cost optimization from the Section 3 objective is intentional for the following reasons:
> 1. **Decoupled Problems**: Communication cost optimization is a direct goal of the dynamic clustering and pruning mechanism, complementing but not interfering with the global framework optimization in Section 3.
> 2. **Logical Clarity**: Placing the communication cost discussion in Section 4.4 provides a focused explanation of its connection to the dynamic mechanism, ensuring clarity for readers.
>
> #### **3. Relationship Between the Federated Learning Objective and Communication Cost Optimization**
> Our method integrates these two objectives by balancing model performance and communication costs through the dynamic clustering and pruning mechanism. Specifically:
>
> 1. **Theoretical Framework**: The optimization objective in Section 3 lays the theoretical foundation for personalized learning by effectively balancing global knowledge and local data characteristics, ensuring robust model performance.
> 2. **Communication Cost Optimization**: Section 4.4 builds on this framework to address communication cost and bandwidth heterogeneity by proposing dynamic pruning and clustering strategies, which reduce communication overhead while maintaining model performance.
>
> By explicitly separating these two aspects, we ensure a clear and logical flow in the paper, while demonstrating how AdFedWCP achieves a practical balance between model performance and communication efficiency.
>
> ---
>
> ### On the Optimization of Figure 1 (Weakness 4):
> We agree that the presentation of Figure 1 can be further optimized to more clearly illustrate the overall framework of AdFedWCP and the calculation process of key metrics. We have refined it in the revised version. The complete AdFedWCP framework diagram can be found in Figure 4 in Appendix G.
>
> ---
>
> ### Supplementary Learning Curves (Weakness 5)
> We have included training convergence curves in the revised paper to provide a clearer comparison of the performance of AdFedWCP and other baseline methods. The curves are presented and analyzed in Appendix E.5.
>
> ---

---

> ### Author Response · Authors · 2024-11-23
> **Response to Weaknesses 6, 7 and 8**
>
> ### Regarding Whether Client Bandwidth Changes Over Time (Weakness 6)
>
> In our experiments, **client bandwidth remains fixed throughout the training process** and does not vary over time. This fixed-bandwidth design was chosen for the following reasons:
>
> 1. **Simplifying Experimental Variables**:
> Using fixed bandwidth allows us to focus on evaluating the core effectiveness of the AdFedWCP method without introducing additional dynamic variables, such as time-dependent bandwidth fluctuations. This avoids potential complexities arising from dynamic bandwidth changes, ensuring a clearer assessment of AdFedWCP's performance.
>
> 2. **Validation of Bandwidth Heterogeneity**:
> Although the bandwidth is statically set, it is heterogeneously distributed across clients. Specifically:
>    - Each client is assigned a different fixed bandwidth to simulate the variability seen in real-world environments.
>    - This static heterogeneity is sufficient to validate AdFedWCP's ability to address bandwidth constraints and perform effectively in heterogeneous environments.
>
> We have provided detailed description about this setting in the reivsed version. The bandwidth setting of the experiment is explained in Section 5.4 of the paper.
>
> ---
>
> ### The limitations of AdFedWCP (Weakness 7):
>
> We have included a discussion of the potential limitations of AdFedWCP in the revised paper to provide a more comprehensive overview of its applicability and areas for improvement. The potential limitations of AdFedWCP are discussed below and included in Appendix F.
>
> While AdFedWCP demonstrates strong performance in addressing bandwidth heterogeneity and improving communication efficiency, there are several limitations that warrant consideration for future research and practical applications. First, AdFedWCP assumes static communication bandwidths for clients throughout the training process. While this simplifies the experimental setup and enables controlled evaluation, real-world federated learning systems often encounter dynamic bandwidth fluctuations. Incorporating mechanisms to address dynamic bandwidth changes could further enhance the robustness and adaptability of AdFedWCP in practical deployments.
> Second, AdFedWCP uses the Imprinting method for layer importance evaluation, which is computationally efficient and well-suited for resource-constrained federated learning environments. However, this approach limits the evaluation to a single method, leaving the potential benefits of other importance evaluation techniques unexplored. Future work could investigate alternative methods, such as saliency-based or gradient-based techniques, to optimize the pruning strategy further and improve model performance.
>
> Lastly, AdFedWCP primarily focuses on addressing bandwidth heterogeneity among clients, with limited consideration for computational heterogeneity, such as differences in processing power or memory capacity across devices. While the weight clustering pruning mechanism has the potential to reduce computational overhead by creating sparse matrices, the sparsity generated by clustering may not be structured. As a result, the improvement with computational efficiency is not as significant as that of structured sparsity methods. Extending AdFedWCP to explicitly address computational heterogeneity or structured sparsity could significantly enhance its applicability in highly resource-constrained environments.
>
> These limitations suggest clear directions for future work, including the development of strategies to handle dynamic bandwidth, exploration of alternative layer importance evaluation methods, and explicit optimization for computational heterogeneity. Addressing these challenges could further enhance the adaptability, scalability, and efficiency of AdFedWCP in diverse and practical federated learning scenarios.
>
> ---
>
> ### On the Improvements to Tables (Table 1 and Table 2) (Weakness 8):
> We have emphasized the best values in bold and underlining the second-best values of Tables in the revised version.

---

> ### Author Response · Authors · 2024-11-23
> **Response to Question 1**
>
> ### Regarding the Hyperparameter $\lambda$ in Lines 130-131 (Question 1)
>
> $\lambda$ is a critical hyperparameter that balances the update weights between the global and local models. Its role is reflected in the following aspects:
>
> 1. **Global Knowledge Sharing**: By incorporating momentum information from the global model, $\lambda$ enables the local model to absorb global knowledge, thereby mitigating biases caused by data distribution heterogeneity.
> 2. **Personalized Learning**: A well-tuned $\lambda$ allows the local model to retain personalized characteristics while maintaining a certain level of consistency with the global model, enhancing the overall performance of the federated learning system.
>
> To accommodate the varying data and model states across different training phases, we implemented a loss-based **annealing mechanism** in our experiments to dynamically adjust the influence of $\lambda$. This mechanism improves the stability and efficiency of training, especially in heterogeneous data environments. However, since it is not a primary innovation or contribution, we did not elaborate on it in the paper.
>
> #### Mathematical Expression of the Annealing Mechanism
> The annealing mechanism dynamically adjusts the decay factor of the momentum coefficient based on the relationship between the current loss and the exponentially smoothed loss. The core formulas are as follows:
>
> 1. **Exponential Smoothing Loss Calculation**:
> $L_{\text{exp}}^{(t)} = \alpha L^{(t)} + (1 - \alpha) L_{\text{exp}}^{(t-1)}$
>
> where:
> - $L^{(t)}$: The current loss at the $t$-th iteration.
> - $\alpha$: A loss balancing coefficient controlling the influence ratio of historical and current losses (set to 0.5 in our experiments).
>
>
> 2. **Momentum Decay Factor Adjustment**:
>
>    $
>    d^{(i)} =
>    \begin{cases}
>    \min(b^{i+1} \times 1.1, 0.8), & \text{if } L^{(t)} < L_{\text{exp}}^{(t)} \\\\
>    \max(b^{i+1} / 1.1, 0.1), & \text{otherwise}
>    \end{cases}
>    $
>
> where:
>    - $i$: Current iteration index.
>    - $b$: Initially set to 0.5 to control the decay rate.
>
>
> 3. **Model Update Rule**:
>   Using the decay factor, the update rule is:
>
>   $\nabla \theta_{i}^{(t)} = \nabla \theta_{i}^{(t)} + d^{(i)} \cdot \Delta \theta_{\text{ref}}$
>
>   where:
>   - $\Delta \theta_{\text{ref}}$: The difference between global and local model parameters.
>
>
>
> #### Influence Analysis of $\lambda$
> ##### **Effects of Fixed Values:**
> - **When $\lambda$ is too large**:
>   - The local model overly relies on global information, potentially losing its adaptability to local data, especially in highly heterogeneous data distributions.
>   - Convergence challenges arise, as observed in experiments where $\lambda = 0.8$, leading to significantly lower accuracy.
>
> - **When $\lambda$ is too small**:
>   - The local model tends to overfit its local data distribution, ignoring the guiding role of global information, thereby compromising the overall performance of the federated model.
>
> ##### **Advantages of Dynamic Adjustment:**
> - During the early training stages, global knowledge sharing is crucial as the model is not yet stable. Increasing the influence of $\lambda$ appropriately can accelerate convergence.
> - In later stages, as the model becomes more stable, local data plays a more significant role. Reducing the influence of $\lambda$ enhances personalization.
>
> #### Experimental Results
> To validate the effectiveness of $\lambda$ and the annealing mechanism, we conducted comparative experiments with different fixed $\lambda$ values and dynamic adjustments.
>
> | Configuration                      | EMNIST Average Accuracy |
> |------------------------------------|--------------------------|
> | $\lambda = 0.1$                    | 82.36%                   |
> | $\lambda = 0.45$                   | 70.26%                   |
> | $\lambda = 0.8$                    | 64.70%                   |
> | Momentum Annealing ($\lambda$ Dynamic Adjustment) | **85.12%**         |
>
> ##### **Analysis**:
>
> 1. Larger $\lambda$ values (e.g., 0.8) caused the model to overly depend on global information. This is particularly detrimental in highly heterogeneous data scenarios, limiting the effectiveness of personalized learning and resulting in poor convergence.
>
> 2. Smaller $\lambda$ values (e.g., 0.1) led to overfitting to local data, failing to effectively integrate global knowledge.
>
> 3. The momentum annealing strategy dynamically adjusted $\lambda$, achieving a balance between global knowledge and local adaptation at different training stages, ultimately delivering the highest accuracy.
>
> ---

---

> ### Author Response · Authors · 2024-11-23
> **Response to Question 1 about $\xi$ and $\zeta$**
>
> ### Regarding the Hyperparameters $\xi$ and $\zeta$ in Lines 366-367
>
> $\xi$ and $\zeta$ are two critical parameters that control the adjustment magnitude when the number of cluster centers is dynamically modified. These parameters play a vital role in determining the optimization lower bound and affect the trade-off between model compression and accuracy.
>
> #### Mathematical Analysis of Their Impact on Optimization Lower Bound
>
> Based on the optimization objective function in the paper, the settings of $\xi$ and $\zeta$ directly influence the value of the optimization lower bound $\eta$ through the following formulation:
>
> $
> \eta =
> \begin{cases}
> 1 - \xi \cdot |\Delta A|, & \text{if } \Delta A > 0 \\
> 1 + \zeta \cdot |\Delta A|, & \text{if } \Delta A < 0
> \end{cases}
> $
>
> where:
> - $\Delta A = A^{(t)} - A^{(t-1)}$: The change in model accuracy between the current and previous rounds.
> - $A^{(t)}$: The model accuracy at the $t$-th round.
>
> #### Impact on Model Optimization:
> - **$\xi$**:
>   - A smaller $\xi$ avoids prematurely reducing the optimization lower bound when the model's performance improves, contributing to model stability.
>   - If $\xi$ is too large, the optimization lower bound shrinks quickly, potentially leading to insufficient feature capture and limiting model convergence performance.
>
> - **$\zeta$**:
>   - A larger $\zeta$ rapidly increases the optimization lower bound when the model's performance deteriorates, increasing the number of cluster centers and helping to restore model performance.
>   - If $\zeta$ is too small, it may fail to effectively increase the number of cluster centers during performance degradation, making it difficult for the model to adapt to heterogeneous environments.
>
> #### Experimental Results Analysis
> | $\xi$ | $\zeta$ | EMNIST Accuracy | Compression Rate |
> |---------|-----------|------------|------------|
> | 0.1     | 1.5       | 85.12%      | 87.54%      |
> | 0.5     | 1.5       | 84.96%      | 87.47%      |
> | 1       | 1.5       | 85.00%      | 87.30%      |
> | 0.1     | 1.0       | 84.96%      | 87.55%      |
> | 0.5     | 1.0       | 84.95%      | 87.60%      |
> | 1       | 1.0       | 84.92%      | 87.59%      |
> | 0.1     | 0.5       | 84.98%      | 87.61%      |
> | 0.5     | 0.5       | 85.02%      | 87.655      |
> | 1       | 0.5       | 84.79%      | 87.66%      |
>
>
> We compared different combinations of $\xi$ and $\zeta$ to evaluate their impact on model accuracy and compression rate. The results are clearly presented in the accompanying figure.
>
> - **Trade-off Between Compression and Accuracy**:
>   - Smaller values of $\xi$ (e.g., 0.1) maintain higher accuracy while achieving favorable compression rates.
>   - Larger values of $\zeta$ (e.g., 1.5) significantly improve the model's performance under degradation scenarios, enhancing adaptability to complex heterogeneous environments.
>
>
> - **Optimal Combination**:
>   The experiments show that $\xi = 0.1$ and $\zeta = 1.5$ represent the optimal configuration, achieving a good balance between accuracy (85.12%) and compression rate (87.60%).
>
> Through these experiments, we have verified that dynamically adjusting $\xi$ and $\zeta$ significantly impacts model performance and communication efficiency. Notably, appropriate parameter combinations effectively balance performance with resource consumption.

---

> ### Author Response · Authors · 2024-11-23
> **Response to Questions 2, 3 and 4**
>
> ### Supplementary Learning Curves (Question 2)
> Please reference our responses to Weakness 5.
>
> ---
>
> ### Regarding Client Participation in Training (Question 3)
> In our experimental setup, **all clients participate in training in every round**. This design choice was based on the following considerations:
> 1. **Fairness**: Ensuring that all client data contributes to the training of the global model helps avoid overlooking certain datasets and enhances the generalization ability of the model across all clients.
> 2. **Simplified Experimental Variables**: By involving all clients in training, we can focus on evaluating the effectiveness of the proposed method (e.g., AdFedWCP) without introducing additional complexity from client selection strategies, such as partial participation or dynamic sampling.
> We have clarified it in the revised version. Lines 467 to 469 of the paper describe the client setup for the experiment.
>
> ---
>
> ### Regarding Whether Client Bandwidth Changes Over Time (Question 4)
> Please reference our respones to Weakness 6.

---

### Comment · Area_Chair_e5T2 · 2024-11-25
**Acknowledge the author responses**

Dear Reviewers,

Thank you very much for your effort. As the discussion period is coming to an end, please acknowledge the author responses and adjust the rating if necessary.

Sincerely,
AC

---

### Comment · Area_Chair_e5T2 · 2024-11-28
**Discussion needed**

Dear Reviewers,

As you are aware, the discussion period has been extended until December 2. Therefore, I strongly urge you to participate in the discussion as soon as possible if you have not yet had the opportunity to read the authors' response and engage in a discussion with them. Thank you very much.

Sincerely,
Area Chair

---

### Meta-Review · Area_Chair_e5T2 · 2024-12-18

**Metareview:**

This paper proposes a personalized federated learning approach, specifically designed to optimize communication efficiency in heterogeneous network environments.  Two reviewers raised serious concerns on presentation (many unclear points) and amount of contribution (confined to communication).  Although the authors provided the additional responses during the discussion period, the paper needs a major revision to incorporate such clarifications.  Thus, I recommend a reject.

**Additional Comments On Reviewer Discussion:**

* Reviewer 61kH increased his/her rating due to the authors' rebuttal. However, he/she mentioned that bandwidth heterogeneity is not defined and analyzed perfectly in the paper.
* The other two reviewers did not respond to the authors' rebuttal.  The authors' responses are too lengthy, and it is hard for me to validate their responses.  The authors' answers are not incorporated into the paper.

---

### Decision · Program_Chairs · 2025-01-22

Reject